# Identifying DNase I hypersensitive sites as driver distal regulatory elements in breast cancer

Matteo D'Antonio [1], Donate Weghorn[2], Agnieszka D'Antonio-Chronowska[3], Florence Coulet[4,10], Katrina M. Olson[5,6], Christopher DeBoever [7], Frauke Drees[4], Angelo Arias[4], Hakan Alakus[4,11], Andrea L. Richardson[8,12], Richard B. Schwab[1,9], Emma K. Farley[5,6], Shamil R. Sunyaev[2] & Kelly A Frazer[1,3,4]

Efforts to identify driver mutations in cancer have largely focused on genes, whereas non-coding sequences remain relatively unexplored. Here we develop a statistical method based on characteristics known to influence local mutation rate and a series of enrichment filters in order to identify distal regulatory elements harboring putative driver mutations in breast cancer. We identify ten DNase I hypersensitive sites that are significantly mutated in breast cancers and associated with the aberrant expression of neighboring genes. A pan-cancer analysis shows that three of these elements are significantly mutated across multiple cancer types and have mutation densities similar to protein-coding driver genes. Functional characterization of the most highly mutated DNase I hypersensitive sites in breast cancer (using in silico and experimental approaches) confirms that they are regulatory elements and affect the expression of cancer genes. Our study suggests that mutations of regulatory elements in tumors likely play an important role in cancer development.

[1] Moores Cancer Center, University of California, La Jolla, San Diego, CA 92093, USA. [2] Division of Genetics, Brigham and Women's Hospital, Harvard Medical School, Boston, MA 02115, USA. [3] Institute for Genomic Medicine, University of California, La Jolla, San Diego, CA 92093, USA. [4] Department of Pediatrics, University of California, La Jolla, San Diego, CA 92093, USA. [5] Department of Medicine, Division of Cardiology, University of California, La Jolla, San Diego, CA 92093, USA. [6] Division of Biological Sciences, Section of Molecular Biology, University of California, La Jolla, San Diego, CA 92093, USA. [7] Bioinformatics and Systems Biology, University of California, La Jolla, San Diego, CA 92093, USA. [8] Department of Pathology, Brigham and Women's Hospital and Harvard Medical School, Boston, MA 02115, USA. [9] Department of Medicine, School of Medicine, University of California, La Jolla, San Diego, CA 92093, USA. [10]Present address: Department of Genetics, Pitie-Salpetriere Hospital, Pierre and Marie Curie University, Paris 75013, France. [11]Present address: Department of General, Visceral and Cancer Surgery, University of Cologne, Cologne 50937, Germany. [12]Present address: The Johns Hopkins University School of Medicine, Baltimore, MD 21205, USA. Matteo D'Antonio and Donate Weghorn contributed equally to this work. Correspondence and requests for materials should be addressed to E.K.F. (email: efarley@ucsd.edu) or to S.R.S. (email: ssunyaev@rics.bwh.harvard.edu) or to K..F. (email: kafrazer@ucsd.edu)

Efforts to identify driver mutations have so far largely focused on coding genes, although there have been several recent analyses focused on non-coding sequences. Weinhold et al.[1] analyzed non-coding functional sequences, identifying 193 regulatory elements with elevated mutation densities, and then characterized in depth driver mutations in the promoters of three genes (SDHD, PLEKHS1 and WDR74). Nik-Zainal et al.[2] analyzed the mutational landscape of 560 breast cancers and also found putative driver mutations in the promoters of PLEKHS1 and WDR74, as well as in the promoter of TBC1D12. Fredriksson et al.[3] focused on regions immediately upstream of transcription start sites, identified 17 recurrent promoter mutations and characterized mutations in the TERT promoter. Melton et al.[4] developed a model to identify driver mutations as outliers of a Poisson distribution. Using this model, they analyzed recurrently mutated positions, detected nine putative driver regulatory elements and experimentally confirmed by reporter assays that the recurrent mutations in three of these elements near TERT, GP6 and BCL11B result in altered expression. A study by Puente et al.[5], which focused on chronic lymphocytic leukemia, identified an enhancer of PAX5 as a putative driver element. Overall, in spite of some recent progress, the analytic approaches for identifying driver regulatory elements are not as developed as those for identifying driver genes and relatively few driver regulatory elements have been characterized.

Breast cancer is a highly heterogeneous disease characterized by four major clinically relevant phenotypes[6] that have specific gene expression patterns[7], but overlapping mutational profiles[8]. Mutational analyses of exomes have shown that combining all breast cancer subtypes together results in increased sensitivity for identifying driver genes[8–10]. In this study, we have analyzed whole genome sequences for 657 breast cancer samples and more than one thousand tumors across 19 additional cancer types to detect non-coding driver mutations. We focused on analyzing DNase I hypersensitive sites (DHSs) as previous pan-cancer analyses of non-coding sequences have shown that regulatory elements associated with DHSs have decreased somatic mutation rates compared with the rest of the non-coding genome, suggesting that mutations in these regions have a driver role in cancer[11–13]. Given the fact that local mutation density is extremely variable across the genome[14, 15], in this study we developed a statistical method that takes into account the influence of DNA sequence characteristics, replication timing, and chromatin on local mutation rates[15]. We then applied a series of enrichment filters resulting in the identification of ten significantly mutated DHSs that are both associated with the aberrant expression of neighboring cancer genes and mutated in two independent sets of breast tumors. A pan-cancer analysis showed that three of the regulatory elements are putative drivers in multiple tumor types. We functionally characterized the four DHSs most highly mutated in breast cancer with a combination of in silico and experimental approaches, including CRISPR and animal models, and confirmed they are regulatory elements of known cancer genes.

## Results

**Mutational landscape in discovery breast cancer samples.** Previous genome-wide mutational analyses of exomes have shown that combining all breast cancer phenotypes together results in increased sensitivity to identify driver genes[8, 10]. In order to detect driver DHSs that are important across all four clinical phenotypes we obtained whole-genome sequences from TCGA for 47 breast cancers (and matched normal samples), representing all four categories (4 HR−/HER2+, 6 HR+/HER2−, 15 HR+/HER2+ and 22 triple negative, Supplementary Fig. 1,

Supplementary Table 1). We investigated the mutational landscape of these tumors to determine if they are similar to previously analyzed breast cancers. Using MuTect[15] we identified 193,958 high confidence somatic mutations corresponding to an average of 4,127 mutations (range 3,241 to 41,714) per patient (Fig. 1a, Supplementary Table 2, Supplementary Data 1). On average, there are 53 mutations (range 10 to 225) per tumor in coding sequences. Examining 12 genes previously reported as harboring driver mutations in breast cancer[8], we identified mutations at the expected density in TP53 (33 mutations in 32 samples, 68.1%) PIK3CA (12 mutations, 25.5%), CDH1 (2 mutations, 4.3%), MLL3 (2 mutations, 4.3%) and GATA3 (one mutation, 2.1%)[8] (Fig. 1b). We did not observe mutations in six genes (CTCF, MAP2K4, PIK3R1, PTEN, RUNX1 and TBX3) known to be recurrently mutated at lower frequency (<5%), or MAP3K1 that is mutated at high frequency (13%) only in luminal A tumors[8]. TP53 has a high incidence of frameshift and nonsense mutations (6 and 4, respectively, Supplementary Data 2), as previously observed[8]. Among non-driver genes mutated at high frequency in our discovery set (>3 samples, 8%), we detected several known to be hypermutated in many cancer types (TTN, MUC17, MUC16, OBSCN), mostly due to their length (>4,000 codons, with long introns often spanning more than 1 Mb) and believed to primarily harbor passenger mutations (Fig. 1c)[14–16]. Examining the somatic substitution frequencies for all samples, the most common substitution is CG–TA (Fig. 1d), comprising 32% of all somatic SNVs, similar to previous observations[17]. Recently, breast cancers were shown to often display kataegis, i.e. localized hypermutation, proposed to result from cytosine deaminations catalyzed by APOBEC proteins[18]. We inspected the distribution of distances between mutations in the discovery samples (Fig. 1e, f) and observed 69 kataegis loci in 29 samples (59.6%, Supplementary Table 3)[19], levels similar to previous reports (~50%)[18]. Overall, these data show that the mutational rates and patterns in the 47 discovery samples are generally consistent with breast cancer samples analyzed in previous genome-wide experiments[16, 17].

**Identifying DHSs significantly mutated in breast cancer.** Several groups have previously shown that the local mutation density is lower in regulatory elements that are active in the cell-type of origin of the tumor compared to regulatory elements in other cell types[11]. This observation supports the hypothesis that mutations in functional genomic regions are usually deleterious and negatively selected. Therefore, increased local mutation density relative to the expected density under a neutral model of evolution is an evidence of positive selection, suggesting that the mutations are functional and hence referred to as drivers[20, 21]. Based on these assumptions, we focused on the two breast cell lines in ENCODE[22], the breast cancer cell line T47-D and a human mammary fibroblast (HMF) line, and derived a list of DHSs. Of note, HMFs are not in the same lineage as the epithelial cells that lead to ductal carcinoma. Analyzing the data of these two cells lines, we obtained 334,781 DHSs corresponding to 118 Mb that do not overlap either RefSeq exons or kataegis loci (Fig. 2a, Supplementary Table 3, Supplementary Data 3). To reduce the number of false negatives, all DHSs (independent of peak height) were included. In the 47 discovery samples, we detected mutations in 14,087 (4.2%) of these 334,781 breast DHSs. To identify breast DHSs enriched for mutations over neutral expectation, we first estimated the expected number of mutations for each DHS assuming absence of selection. To take into account known factors responsible for the mutation rate heterogeneity, we grouped the 334,781 breast DHSs into 223 clusters on the basis of chromatin and DNA sequence

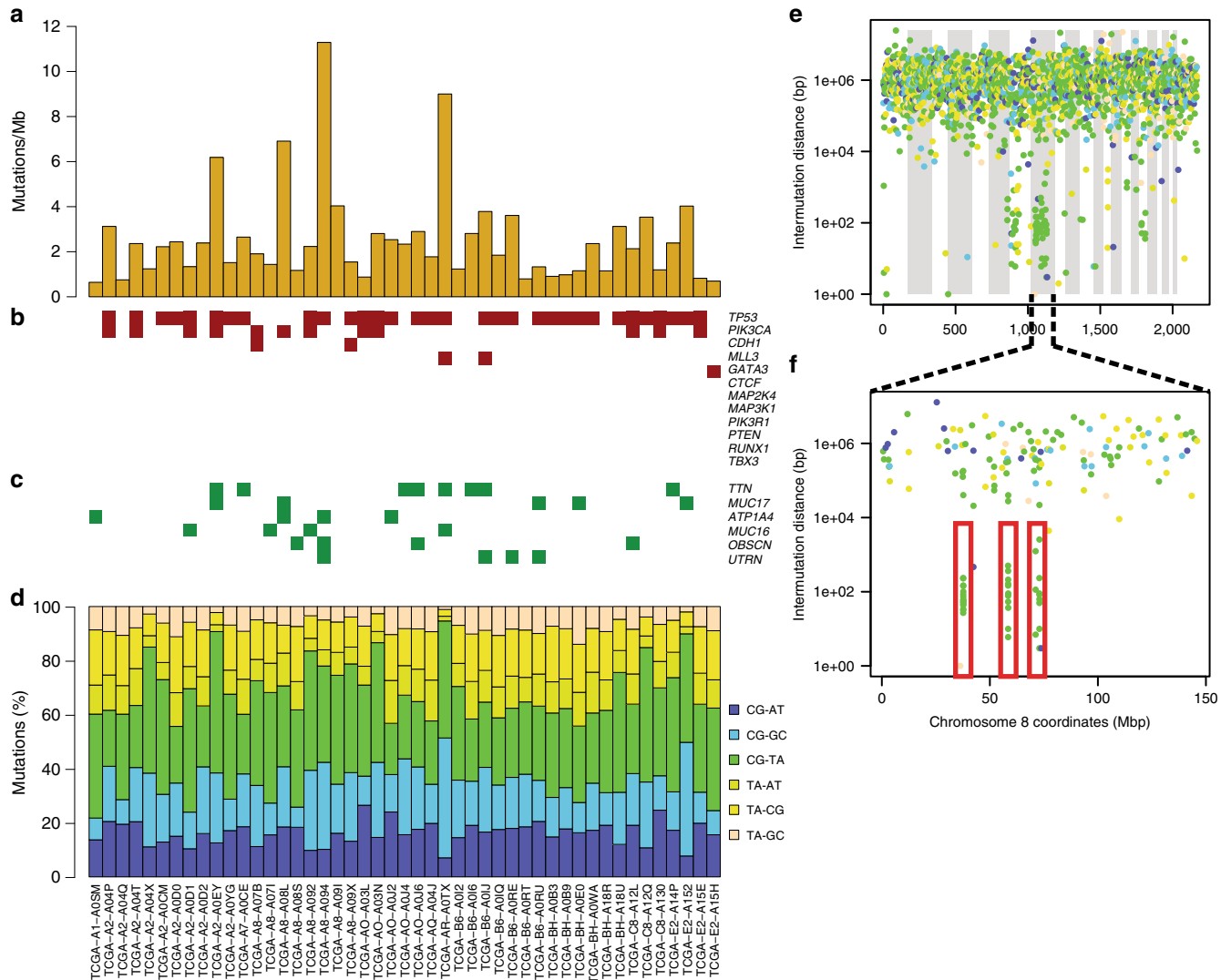

**Fig. 1** The 47 discovery breast tumors have typical mutational profiles. For each sample (sample IDs at bottom of panel 1D), we show **a** the number of mutations per Mb; **b** if a mutation is present in one of the 12 known breast cancer driver genes or **c** in non-driver genes that are mutated in at least 3 (8%) discovery samples; and **d** genome-wide SNV substitution frequencies; CG–TA mutations are the most common. **e** The mutational profile in discovery sample TCGA-A2-A0D1 (SNV substitution types colored-coded as in panel 1D). The X-axis shows all mutations ordered by mutation index (from the first mutated position on chromosome 1 to last mutated position on chromosome X) and the Y-axis represents the average distance between each mutation and its two neighboring mutations, in log-scale. **f** Shows three kataegis loci (*red rectangles*) on chromosome 8

characteristics known to affect mutation frequency: (1) DNA replication timing[23], (2) open and closed chromatin status[24], (3) GC content, (4) local gene density[15], and (5) expected mutations based on trinucleotide composition (Supplementary Data 4). For each of the 223 clusters of DHSs with similar expected mutation densities, we inferred the background mutation probability. Then, we calculated the probability $p$ for each DHS of having the observed or a higher number of mutations assuming Poisson distribution (see Methods section), and we used this $p$-value as our test statistic. We estimated its expected distribution for our dataset under a model of neutral evolution by simulating mutations (random expectation, Fig. 2b, Supplementary Fig. 2). Based on the latter expected distribution of $p$, we defined a false discovery rate (FDR) as the expected fraction of sites that have a value $p$ smaller than the FDR threshold $q$ just by chance. We considered the value of $p$ at which the FDR is 0.25 as the threshold for significance ($p < 0.00171$ for breast DHSs Fig. 2c). Using this threshold, we identified 637 potentially significantly mutated breast DHSs (Filter 1, Fig. 2a, b).

In a second step of our significance analysis, we compared the results of the test statistic $p$ for the mutations in breast DHSs to those derived from analyzing mutations in DHSs from 13 different control tissues (Supplementary Tables 4 and 5). A priori we expect to see no selection signal in the control tissues and, hence, no differences between the distribution of $p$ in the control tissues and in the simulated random data, which correspond to our model of neutral evolution. We clustered the DHSs for each of the 13 control tissues in the same manner as in breast and likewise calculated $p$ for each DHS in control tissues. Because control tissue DHSs serve as a proxy for neutrally evolving sites, we first tested whether the values $p$ in the 2,681 individual DHS clusters for the 13 control tissues are compatible with the random expectation from our Poisson model, using a Kolmogorov–Smirnov test (Supplementary Table 5). We found that mutations in DHS clusters of the control tissues agree with the expectation under the Poisson model of neutral evolution. Conversely, this does not apply to the distribution of $p$ in breast, suggesting that driver mutations are likely present in the set of

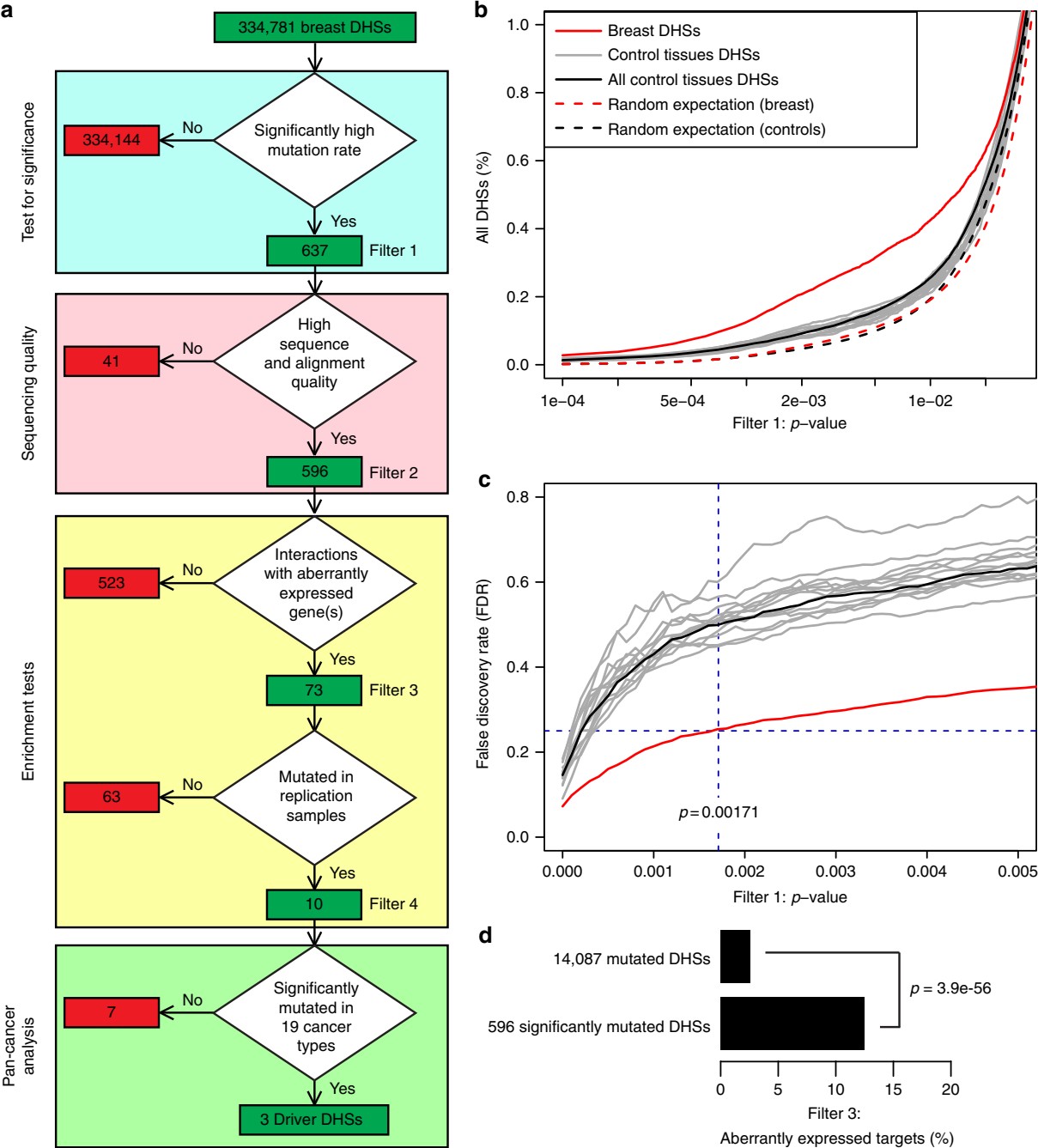

**Fig. 2** Identification of driver DHSs in breast cancer. **a** Analysis flowchart for identifying putative driver DHSs. The 334,781 breast DHSs were filtered consecutively for: (1) a significantly high mutation density ($p < 0.00171$, corresponding to FDR < 0.25, see Methods); (2) high sequence and alignment quality; (3) having a known target gene within 500 kb that is aberrantly expressed in breast tumors; (4) being mutated in two replication sets of breast cancer samples; and (5) being significantly mutated above background in a pan-cancer analysis. **b** Cumulative distributions of test statistic $p$ used in Filter 1 that are <0.05 (log-scale) for: (1) breast DHSs; (2) the 13 control tissues DHSs (*one gray line* for each); (3) "All control tissues" DHSs represent the union of DHSs that are active in the 13 control tissues; (4) and (5) simulated mutations (random expectation) in breast and control tissues DHSs, respectively. (**c**) False discovery rate and $p$ threshold. We set the threshold to a value $p*$, below which DHSs are considered significant, according to Eq. 4, limiting the expected false discovery rate to 0.25. We applied this FDR threshold to breast (*red*) as well as to each control tissue's DHSs (*gray*) and all control tissues DHSs (representing the union of all control tissues DHSs, *black*). The *vertical dashed line* represents $p*$ for breast DHSs ($p* = 0.00171$). **d** The 596 mutated DHSs passing Filter 2 are enriched for DHSs associated with target genes that are aberrantly expressed

breast DHSs, resulting in deviations from the expected distribution. Similarly, for control tissues, the fraction of DHSs passing the FDR = 0.25 filter is significantly lower than in breast ($p$-value $= 1.7 \times 10^{-113}$, Fisher's exact test, Supplementary Fig. 3a), because the values $p$ at this FDR threshold are substantially smaller (maximum is $p = 0.00047$ Supplementary Table 5). This is due to

the fact that control tissue DHSs have a lower signal-to-noise ratio (where "signal" is intended to denote driver DHSs and "noise" is the random expectation, corresponding to the assumption of neutral evolution) than breast DHSs. If DHSs in the control tissues were all true negatives (i.e., they do not harbor any driver mutations), the distribution of $p$ values would be the

same as in the random expectation and the FDR would be equal to one for all *p*. Indeed, we observe a significantly lower signal than in breast DHSs (Fig. 2c), albeit still positive, suggesting that there are DHSs active in the control tissues that may be positively selected for mutations in breast cancer. There are several possible reasons for this including: (1) some of these DHSs are active in multiple control tissues, and a fraction of them are likely false negatives in the set of breast DHSs in ENCODE; (2) some of the significantly mutated DHSs active in control tissues may be false positives, due to the fact that we used a loose FDR threshold (0.25); (3) these DHSs may be a different class of regulatory elements in breast that do not bind DNA-binding proteins, and therefore are not detected as DHSs; and (4) the genomic regions corresponding to these DHSs may be inactive in the two breast samples that we used to define breast DHSs (HMF and T-47D), but may be active in other breast cancer samples.

**Enrichment filters identify putative driver DHSs.** As has been shown in whole exome analyses to identify novel cancer genes, statistical methods that select for sequences mutated at higher frequency than expected are not sufficient to detect driver mutations and often result in the detection of false positives, such as genes like *TTN* or *MUC17*, that are neither expressed in the tumor nor in its associated normal tissue[14–16]. Given our use of an FDR = 0.25 for Filter 1, the 637 significantly mutated breast DHSs may have a high number of false positives. Therefore, to eliminate false positives and enrich for a set of DHSs with characteristics expected of driver regulatory elements, such as functional activity in associated tissue, we applied additional filters to the 637 breast DHSs. To eliminate elements significantly enriched for mutations due to sequencing or alignment issues, we filtered out DHSs with poor sequencing quality (mean read count, mean read quality) or alignments (fraction of improperly aligned reads) compared to the sequencing quality distribution of the DHSs in their associated clusters (Filter 2). We eliminated 33 DHSs whose mean read count and/or mean read quality was lower than 2 standard deviations from the cluster mean (*Z*-score < −2) and 8 DHSs whose fraction of improperly aligned reads was higher than 2 standard deviations from the cluster mean (*Z*-score > 2, Supplementary Data 5). We filtered the remaining 596 significantly mutated DHSs to enrich for elements that: (1) are associated in the tumor(s) in which the DHS is mutated with the aberrant expression of gene(s) located within 500 kb; (2) have established evidence of physical interactions with the aberrantly expressed gene(s) in the ENCODE[25] and/or 4D Genome datasets[26]; and (3) the associated aberrantly expressed gene(s) do not overlap somatic copy number variations. We identified 73 DHSs that fulfill these three criteria (Filter 3, Fig. 2a, Supplementary Data 6). Of note, although mutations in these DHSs are associated with aberrant expression of the target genes, this is not evidence of causality and thus we applied additional filters and conducted functional experiments as described below.

To assess the effectiveness of the significance test (Filter 1) to identify functional mutations, we examined how it enriches for DHSs associated with aberrantly expressed genes compared with all mutated DHSs. For the 14,087 mutated DHSs, we determined that 402 (2.9%) have known interactions with a gene that is within 500 kb and aberrantly expressed (Fig. 2d). Of the 596 significantly mutated DHSs (Filter 2), we detected 73 (12.2%, Filter 3) associated with one or more aberrantly expressed genes (*p*-value = $3.9 \times 10^{-56}$, $\chi^2$ test). Interestingly, the DHSs in the control tissues that pass Filter 1 are also enriched for association with aberrantly expressed genes (albeit significantly lower than the rate of breast DHSs, *p*-value = $2.2 \times 10^{-9}$, Fisher's

exact test), suggesting that the reason these DHSs do not behave as null is because some of them are under positive selection in the breast tumor (Supplementary Fig. 3b). These results show that our significance test (Filter 1) greatly enriches for DHSs that are associated with aberrant gene expression in comparison with two distinct control sets: (1) breast DHSs that do not have significantly high mutation rate (Fig. 2d); and (2) DHSs that are active in other tissues but not in breast (Supplementary Fig. 3b).

To detect putative driver DHSs, we further filtered the 73 breast DHSs based on whether or not they were mutated in two replication sets of breast cancer samples (Filter 4 in Fig. 2a) representing all four clinical phenotypes. In the first replication set, we examined the corresponding sequences in 50 TCGA breast tumors and for a subset of the DHSs performed targeted sequencing in 135 breast cancer samples (Supplementary Fig. 1). We determined that the replication and discovery TCGA samples have similar overall mutation densities (Supplementary Table 6, Supplementary Data 7). We then examined the exons of the 12 breast cancer genes discussed above and showed that this first set of 185 replication breast cancer samples have the expected mutational landscape (Supplementary Fig. 4, Supplementary Data 8). Analysis of the 73 breast DHSs revealed that 16 carried one or more mutations (total of 31 mutations) in these 185 replication samples (Fig. 3a, Supplementary Data 9). DHSs in control tissues associated with aberrantly expressed genes (passing Filter 3) are mutated at a substantially lower frequency than breast DHSs in this first set of replication samples (Supplementary Fig. 3c), confirming that Filter 4 results in the enrichment for likely true driver regulatory elements. These 16 DHSs were further examined in a second replication set of 560 breast tumors with Whole Genome Shotgun (WGS) data (BRCA-EU)[2], of which ten DHSs harbored mutations (range: 1 to 13 mutations) (Supplementary Data 9). As determined in Filter 3 in Fig. 2a, the ten breast cancer DHSs with mutations in both sets of replication samples, which we refer to as putative driver DHSs hereafter, are associated with the aberrant expression of 27 genes (20 overexpressed and 7 downregulated), of which 18 have known roles in cancer (Supplementary Table 7). Interestingly, one gene (*TRIM27*) is an oncogene included in the Cancer Gene Census, six of these genes (*ACSBG1*, *COL20A1*, *DVL1*, *LPCAT1*, *VWA1*, and *ZNF596*) are aberrantly expressed in multiple cancer types and eight (*ACSBG1*, *ATAD3B*, *CLPTM1L*, *COL20A1*, *LPCAT1*, *MAST2*, *RAD54L* and *ZNF596*) are involved in breast cancer. These observations suggest that the ten putative driver DHSs regulate genes whose aberrant expression is associated with cancer.

**Putative driver DHSs are mutated in multiple cancer types.** To determine whether the ten putative driver DHSs that we identified in multiple collections of breast cancer (Fig. 2) may play a role in other tumors, we examined their sequences in 1,097 TCGA cancer genomes from 19 tumor types (Fig. 3a, b). We used MuTect to call somatic mutations and identified 298, corresponding to an average of 29.8 (range: zero to 105) per DHS. To assess our power to detect somatic mutations in a pan-cancer analysis of thousands of cancer samples, we called variants in ten genes known to be highly mutated in multiple cancer types[27], and found on average 50 (range: 14 to 182) mutations per gene (Supplementary Fig. 5), suggesting that our approach has high sensitivity to detect mutations. These results also show that the mutation densities of the ten putative driver DHSs are similar to known cancer genes (Fig. 3b).

We performed two analyses to determine if any of the breast cancer putative driver DHSs are mutated above background in

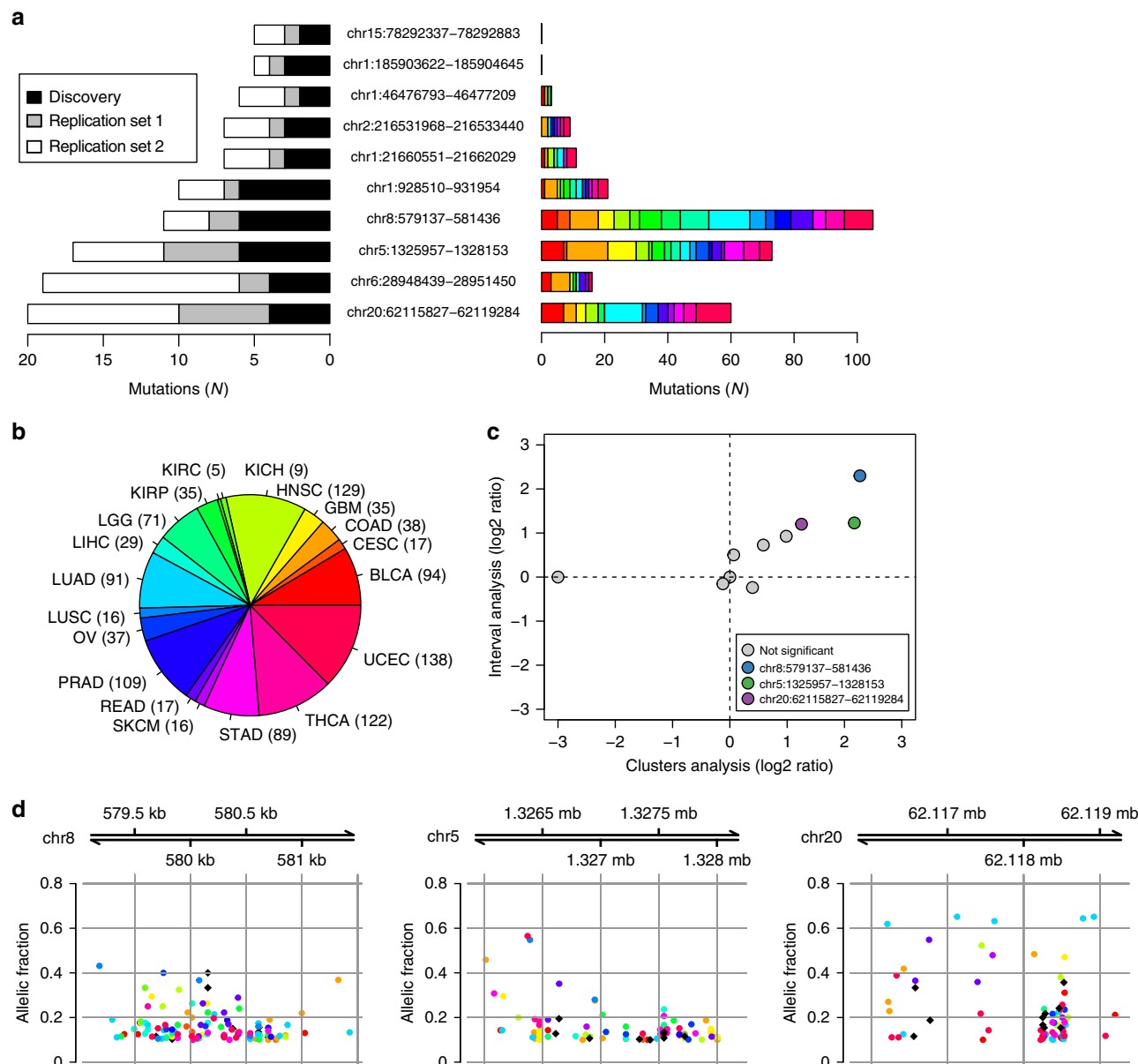

**Fig. 3** Analysis of breast cancer putative driver DHSs in other cancer types. For each of the ten putative driver DHSs, the barplots show the number of mutations detected in **a** discovery and the two replication sets and **b** all 19 cancer types. The pie chart represents the distribution of the 1,097 samples in TCGA across the cancer types (definitions shown in Supplementary Fig. 5). **c** Pan-cancer mutation analysis of the ten putative driver DHSs. The X-axis represents the log2 ratio of the cluster analysis (mutation density of each putative driver DHS compared to 20 random DHSs with similar sequence characteristics). The Y-axis represents the log2 ratio of the interval analysis (each putative driver DHS compared to the surrounding 100 kb). DHSs that are not significant (*gray*) and significant in both tests (color-coded) are shown. **d** Spatial distribution of the mutations within the three putative driver DHSs across tumor types (colored as in **b**; breast cancer is represented by *black diamonds*). The Y-axis represents the allelic frequency of each mutation

other cancer types (Fig. 3c). First, we conducted an interval analysis by comparing the mutation density of each putative driver DHS with that of the surrounding sequence (50 kb upstream and downstream), and found four DHSs (40%) significantly more mutated (Supplementary Data 10). Second, we conducted a clusters analysis by comparing the mutation density of each DHS with that of 20 DHSs randomly chosen from the same cluster (based on GC content, open chromatin, DNA replication time, local gene density and expected context-dependent mutations, Supplementary Data 4), and found three DHSs significantly more mutated, all of which were also found by the first approach (Supplementary Table 8). Of these three DHSs,

the most mutated one, chr8:579137-581436, is significantly mutated in 12 cancer types individually in addition to the pan-cancer analysis, while the other two DHSs, chr5:1325957-1328153 and chr20:62115827-62119284, are respectively significantly mutated in six and seven of the cancer types individually. Mutations in two of these DHSs (chr8:579137-581436 and chr5:1325957-1328153) are distributed across the entire elements, whereas the DHS on chr20:62115827-62119284 has the majority of its mutations concentrated in a 500-bp region next to its 3′ end (45 of its 68 mutations, 66.1%, Fig. 3d), but the mutations are not recurring at single base pair sites. Overall, this pan-cancer analysis showed that three of the breast cancer putative driver

DHSs are mutated above background in multiple tumors at mutation rates similar to driver genes, suggesting that they may be involved in tumorigenesis in several cancer types.

**DHS mutations are associated with *TERT* overexpression**. The most highly mutated DHS in breast cancers is at chr5:1325957-1328153, located within an intron of *CLPTM1L* and 30 kb upstream of *TERT* (Fig. 4a). This putative driver DHS is an enhancer based on ChromHMM, harbors 17 mutations in 13 breast cancer samples (three discovery, four in replication set 1 and six in replication set 2), and when mutated results in overexpression of six genes (Fig. 4b). Four of these genes have known associations with cancer: *TERT* overexpression is associated with glioblastoma, hepatocellular carcinoma and melanoma[1]; *CLPTM1L* inherited variants are associated with lung and pancreatic cancer risk[28]; overexpression of *TRIP13* drives DNA damage in head and neck cancer[29]; and *LPCAT1* overexpression correlates with tumor progression and prognosis in breast, colorectal and prostate cancer[30]. All these genes are located within the same topologically-associated domain (TAD) as the putative driver DHS[31] and three of them (*TERT*, *CLPTM1L* and *LPCAT1*) are also included in the same chromatin loops[32], supporting our observations that the putative driver DHS likely regulates their expression. Since *TERT* overexpression is often associated with mutations in its promoter[33], we examined the *TERT* promoter in the 1,097 TCGA cancer genomes (see Methods) and found 51 mutations, of which 49 are in two loci previously described as recurrently mutated[33] (Supplementary Data 11). The majority of these 51 mutations are in glioblastoma (23) and low-grade glioma (10), with the remaining distributed across bladder, head and neck, lung, melanoma and thyroid tumors. Our findings show that the known driver mutations in the *TERT* promoter do not commonly occur in breast cancer, but *TERT* is frequently overexpressed in breast tumors and this altered expression is associated with mutations in the putative driver DHS (chr5:1325957-1328153). Furthermore, our data suggest that mutations in this putative driver DHS may contribute to the development of cancer by driving overexpression of several different genes.

**Regulatory potential is affected by DHS mutations**. Mutations in the second most highly mutated putative driver in breast cancer, DHS chr6:28948439-28951450 (Fig. 3a), were associated in Filter 3 with the overexpression of a known cancer gene (*TRIM27*), a member of the tripartite motif (TRIM) family of E3 ubiquitin-protein ligases that is involved in several cancer types through interactions with RARα, RB, p300, ERBB2, RET and JUN[34]. To determine if the mutations we identified in this DHS have a functional impact on gene expression, and therefore may be driver mutations, we investigated the effects of four mutations in vivo, using *Ciona intestinalis* as a model system (Supplementary Table 9). This urochordate is an excellent system to use for screening of regulatory variants, because it shares a large part of its transcriptional machinery with higher eukaryotes[35]. The four mutations are distributed across the element with two affecting GATA binding sites and two affecting ETS binding sites Fig. 5a). To determine their effects, we built reporter constructs containing either the wild-type or mutated DHS attached to a minimal promoter[36] and GFP. These constructions were electroporated into *C. intestinalis* fertilized eggs to assay their function. The two GATA mutations result in significant differential expression, with one resulting in overexpression and one in downregulation ($p$-value = $1.25 \times 10^{-12}$ and $p$-value = 0.0032, respectively Fisher's exact test, Fig. 5b–g, Supplementary Table 10): chr6:28950885A>G results in

a seven-fold increased GFP signal in epidermis, whereas chr6:28949254A>C reduces the enhancer activity by 50% in anterior neural plate (a6.5 lineage). We also observed a significant decrease in the enhancer activity for chr6:28950050G>A in multiple tissues ($p$-value = 0.0077 in endoderm, $p$-value = 0.065 in anterior neural plate and $p$-value = 0.015 in secondary notochord, Fisher's exact test, Fig. 5b–g, Supplementary Table 10), whereas chr6:28950040C>T does not result in any significant change. These results demonstrate that three of the four mutations tested result in the aberrant activity of DHS chr6:28948439-28951450, providing evidence that these were likely driver mutations in the cancers in which they were detected.

**DHS deletions alter chromatin structure and gene expression**. The third and fourth most highly mutated putative driver DHSs in breast cancer are at chr8:579137-581436, which was associated in Filter 3 with *ZNF596* downregulation, a gene frequently downregulated in breast cancer[37] and osteosarcoma[38]; and at chr20:62115827-62119284 which was associated with the overexpression of five genes, including *ARFGAP1*, a gene involved in microsatellite instability oncogenesis[39], and *COL20A1*, whose expression levels are used in predictive models for breast cancer risk[40]. To functionally characterize these putative driver DHSs and investigate causality underlying these associations, we used CRISPR to delete the intervals harboring the elements and analyzed changes in gene expression and chromatin accessibility. We chose this approach because it has been shown that deletions of distal regulatory elements result in altered expression of their target genes[41, 42].

We deleted a 2.3 kb interval harboring driver DHS (chr8:579137-581436) in the HEK293T cell line (Fig. 6a). The HEK293T cell line was chosen because: (1) it is easy to transfect; (2) the putative driver DHS is an active regulatory element (Fig. 6b); and (3) the genes identified as targets of the putative driver DHS (Supplementary Data 6) are expressed. We investigated the effects of this deletion on chromatin configuration (ATAC-seq) and gene expression (RNA-seq) in the 2 Mb (+/−1 Mb) interval surrounding the putative driver DHS. Deletion of the putative driver DHS resulted in several large chromatin accessibility changes including gain of two ATAC-seq peaks (3 kb and 410 kb distal) coupled with loss of an ATAC-seq peak 40 kb distal (Fig. 6c–f). Additionally, we observed a significant decreased accessibility to the promoter regions of four nearby genes (*ARHGEF10*, *FBXO25*, *TDRP* and *ZNF596*) and a significant decrease in expression levels of four protein-coding genes (*DLGAP2*, *ARHGEF10*, *TDRP* and *MYOM2*) and four non protein-coding genes (*RP11-91J19.4*, *RPL23AP53*, *CTD-2336O2.1*, and *RP11-439C15.4*) (Fig. 6g–k). *ERICH1*, whose intron harbors the putative driver DHS, is not downregulated and retains the ATAC-seq peak at its promoter when the putative driver DHS is deleted (Fig. 6d). These results show that deletion of DHS chr8:579137-581436 alters the open chromatin configuration of the promoter of the target gene identified in Filter 3 (*ZNF596*), and suggests that the driver mutations in the DHS may affect the expression of additional genes.

We deleted a 901-bp region harboring the portion of the driver DHS chr20:62115827-62119284 that contains the vast majority of its mutations (Fig. 3d) in the HEK293T cell line and investigated expression changes of neighboring genes (Supplementary Fig. 6a). This DHS is located in a gene-dense genomic region (60 genes within 2 Mb distance), of which 15 are downregulated when it is deleted (Supplementary Fig. 6b). Two of these downregulated genes (*ARFGAP1* and *GMEB2*) were identified in Filter 3 as having both known interactions and altered expression associated with mutations in DHS chr20:62115827-62119284. An additional

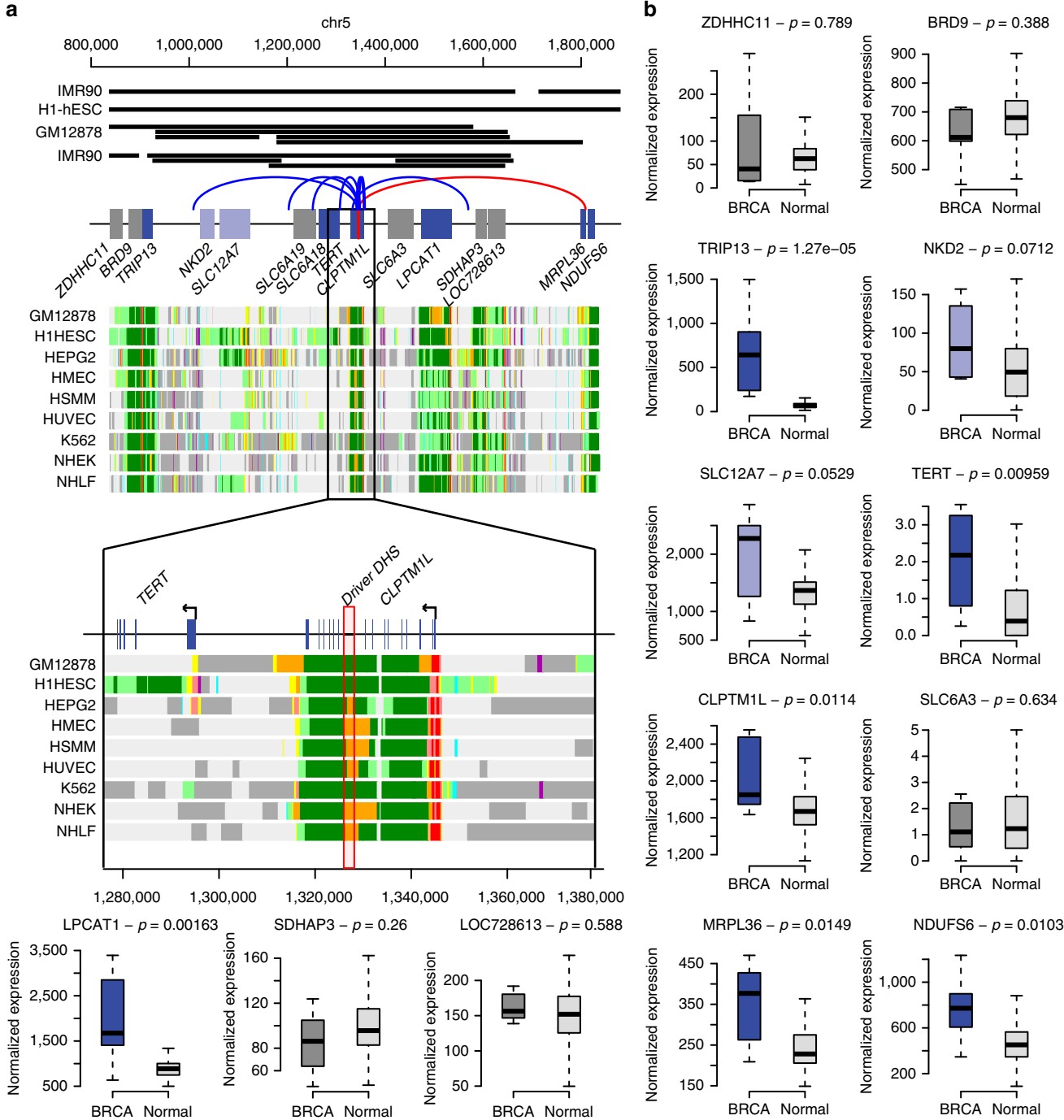

**Fig. 4** Mutations in putative driver DHS chr5:1325957-1328153 result in overexpression of six neighboring genes. Mutations in putative driver DHS chr5:1325957-1328153 result in overexpression of six neighboring genes. **a** The 1 Mb interval surrounding the putative driver DHS (*red rectangle, lower panel*), showing all genes that are expressed in breast cancer. Chromatin states of nine cell lines show that the DHS is an enhancer and that many of the genes in this interval are expressed across diverse tissue types (data derived from the Broad ChromHMM track in the UCSC genome browser[70]; *light green* = weakly transcribed; *dark green* = transcribed, *orange* = strong enhancers, *yellow* = weak enhancers, *red* = active promoters, *light red* = weak promoters, *violet* = inactive/poised promoters, *blue* = distal CTCF/insulators, *dark gray* = polycomb repressed, *light gray* = heterochromatin). *Black lines* show TADs detected in IMR90 and H1-hESC cell lines[31], while *gray lines* show chromatin loops derived from IMR90 and GM12878 cell lines[32]. *Curved lines* show validated (*blue*) and predicted interactions (*red*) between the putative driver DHS and target genes[25, 26]. **b** Boxplots showing expression levels of 13 genes in the seven TCGA breast cancer samples (three discovery and four replication) that harbor mutations in the putative driver DHS (breast invasive carcinoma: BRCA) and in 106 unrelated normal TCGA breast tissues (normal). Significantly overexpressed genes (*dark blue*, *p*-value < 0.05) and those with a trend towards overexpression (*light blue*, *p*-value < 0.1) are indicated. *p*-values were calculated with Wilcoxon test. Boxplots were created with the *boxplot* function in R: *center line* represents the median value, box limits correspond to first and third quartile, while whiskers extend between the minimum and maximum value

downregulated gene, *PPDPF*, had altered expression associated with mutations in DHS chr20:62115827-62119284 in Filter 3 but did not have known interactions. These analyses confirm that DHS chr20:62115827-62119284 has regulatory properties and affect the transcription of the target genes identified in Filter 3 (Fig. 2a).

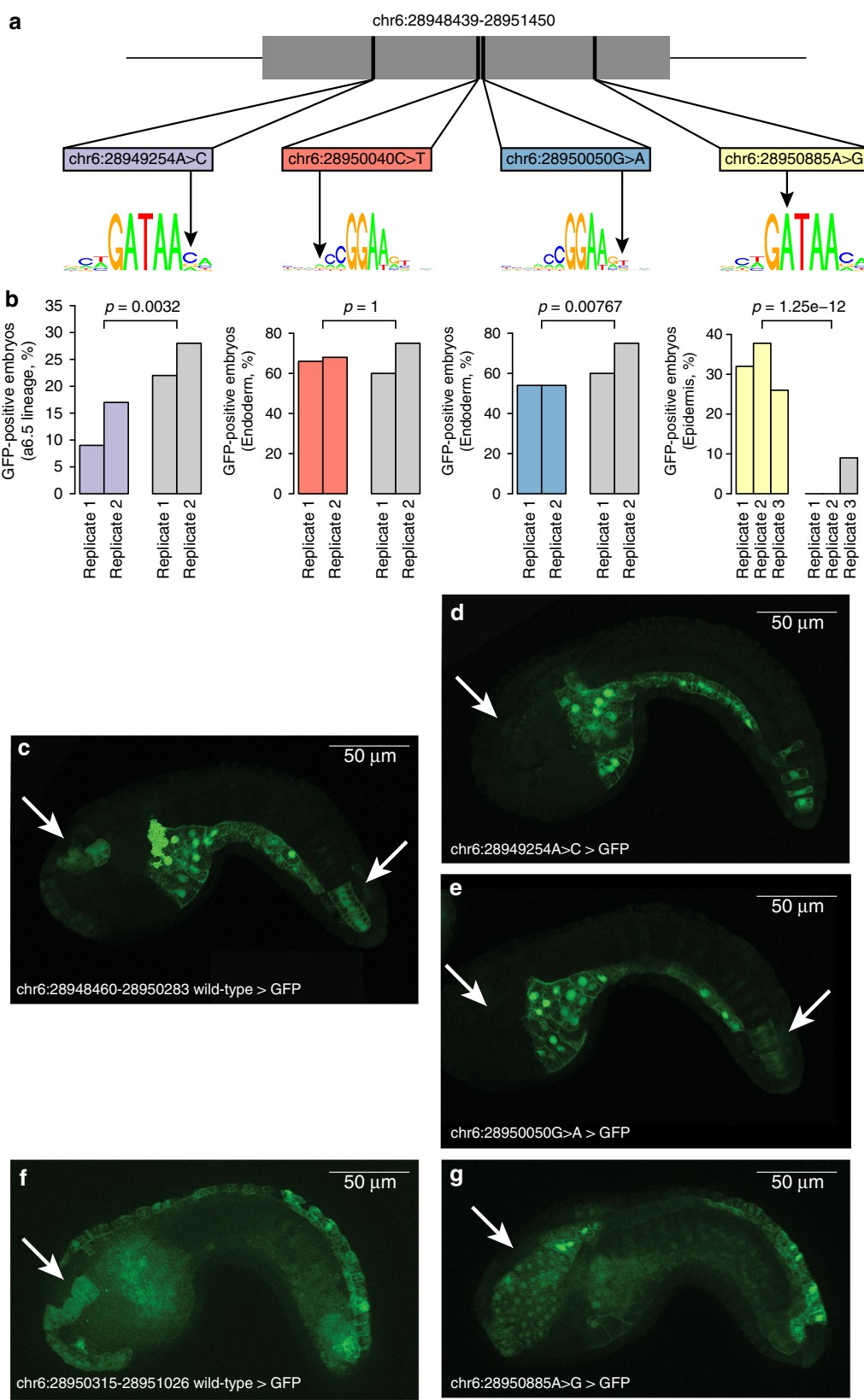

## Discussion

To identify putative breast cancer driver mutations in regulatory elements, we developed a novel method by adapting approaches that have been successfully used to identify recurrently mutated driver genes[14, 16]. We limited the search space to functional sequences comprising 4% of the human genome by focusing on breast DHSs that have reduced local mutation densities similar to coding sequences[11]. As we previously showed that chromatin features in the tissue of origin of a tumor are strongly associated with the tumor's somatic mutation profile[11], we analyzed DHSs that are active in breast, derived from the breast cancer T47-D and HMF cell lines (of note, the mammary fibroblasts are not in the same lineage as the epithelial cells that lead to breast invasive carcinoma). We combined all breast cancer subtypes together because previous studies have shown that this solution results in increased sensitivity for identifying driver genes[8–10]. However, analyzing the different subtypes separately may result in the detection of additional subtype-specific drivers. To detect elements mutated above background, we developed a statistical test, which takes into account genomic features known to influence local mutation densities[15]. To enrich for coding driver mutations, cancer studies frequently only consider functional mutations resulting in non-synonymous amino acid changes, and genes that are mutated above background rate are examined in an independent replication sample set as part of a prevalence screen[21]. Here we applied similar enrichment filters by only retaining significantly mutated DHSs that are associated with aberrantly expressed target gene(s) and mutated in two replication sets of breast cancer samples as putative driver DHSs. We further examined putative breast cancer driver DHSs in a pan-cancer analysis using TCGA tumors across 19 additional cancer types. Our strategy resulted in the identification of three putative driver DHSs across multiple cancer types that have mutation densities similar to the most highly mutated genes previously reported as harboring driver mutations.

We performed a variety of functional validations on the four most highly mutated DHSs in breast cancer. These studies enabled us to determine that the most highly mutated putative driver DHSs is a long-distant regulatory element likely affecting the expression of *TERT* in breast cancer. We used an animal model system to investigate the second most highly mutated putative driver DHS in breast cancer and showed that mutations that alter TFBSs in this element result in altered regulatory activity. For the third and fourth most mutated putative driver DHSs, we showed that their deletion resulted in widespread epigenetic and expression changes of neighboring genes, some of which are known to play important roles in cancer. Although we did not experimentally investigate the other six putative breast cancer driver DHSs, the fact that mutations in many of them are associated with altered expression of genes with known roles in breast cancer suggests that they may be breast cancer-specific drivers.

Our findings in breast cancer do not overlap the results of previous studies that have detected non-coding driver mutations for several reasons, including our use of DHSs as a set of regulatory elements, our application of a novel test statistic and enrichment filters, and the fact that we focused on regions (and not specific base pairs) that are significantly recurrently mutated. We selected breast DHSs because they encompass all regulatory elements that bind DNA-binding proteins, while at the same time limiting the search space to less than 100 Mb. Our test statistic demonstrates that the local mutation probability of DHSs can be inferred from clustering of sites based on similar sequence characteristics, and the resulting mutation counts follow a Poisson distribution. Outliers are enriched for being associated with altered expression of target genes. While previous studies examined only a small fraction of their significantly mutated regulatory elements for effects on gene expression, we have taken advantage of TCGA transcriptome data of the 47 discovery breast tumors as well as 106 normal breasts to examine all 596 significantly mutated DHSs for effects on the expression of known target genes. Additionally, previous studies have focused on specific base pairs that are recurrently mutated due to the fact that the functions of non-coding sequences are still largely unknown and only motifs that are targets of transcription factors are readily characterized[4]. For instance, oncogenic mutations in the *TERT* promoter localize to two specific positions and induce de novo generation of binding sites for ETS transcription factors that result in *TERT* overexpression[3, 4, 33], while loss-of-function mutations in the *SDHD* promoter localize to three different positions and disrupt ELF1 binding sites resulting in down-regulation of *SDHD*[1]. Conversely, the functional consequences of driver mutations in the *PLEKHS1* and *WDR74* promoters are not known. Driver mutations in the *PLEKHS1* promoter result in downregulation of gene expression; although localized at two distinct positions they do not affect transcription factor binding sites but possibly cause an atypical secondary structure in single stranded DNA[1, 4]. Driver mutations in the *WDR74* promoter are broadly distributed across multiple positions and do not seem to affect expression[1, 2]. These later studies combined with our findings suggest that driver mutations can alter the activity of a regulatory element without affecting transcription factor binding sites. Further investigations are necessary to understand what causes these constraints and how mutations in driver DHSs influence the expression of target genes.

## Methods

**Sample collection.** Whole genome sequence data (BAM files) from 97 breast-invasive carcinoma samples (47 as discovery screening and 50 as replication set) and matched normal blood present in TCGA were downloaded from the Cancer Genomics Hub (CGHub, https://browser.cghub.ucsc.edu/, frozen on December 19th 2013)[43]. Similarly to previous analyses[3], this set included only samples with high-coverage whole genome sequencing data (BAM files > 75 Gb). For targeted sequencing of a subset of the samples in replication set 1, 135 breast tumors (of known estrogen receptor (ER), progesterone receptor (PR) and human epidermal growth factor receptor 2 (HER2) status, Supplementary Fig. 1) and matched blood were analyzed (40 from the Cancer Center Biorepository at the University of California, San Diego[44], 16 from the University of California, Irvine[44] and 79 from the Dana Farber Cancer Institute). Informed consent was obtained for all subjects[44] and this collection was approved by the Institutional Review Boards of the University of California at San Diego and of the Dana Farber Cancer Institute.

DNA was isolated from tumor and blood samples[44]: snap-frozen tissue samples were mechanically pulverized, then underwent tissue disruption in lysis buffer and

---

**Fig. 5** In vivo validation of putative driver mutations in DHS chr6:28948439-28951450. **a** Shown are the locations of the four mutations in GATA and ETS transcription factor binding sites that were tested for function. **b** Barplots showing downregulation in anterior neural plate (a6.5 lineage, chr6:28949254 A > C), no change in endoderm (chr6:28950040 C > T), downregulation in endoderm (chr6:28950050 G > A) and overexpression in head epidermis (chr6:28950885 A > G). *p*-values were calculated using Fisher's exact test. (**c**–**g**) Images showing tailbud stage *C. intestinalis* embryos (8 h post fertilization at 21 degrees Celsius) electroporated with (**c**, **f**) indicated reference enhancer > GFP or (**d**, **e**, **g**) indicated mutated enhancer > GFP. **d** Decreased expression in the anterior neural plate is observed when the enhancer is mutated (chr6:28949254 A > C); **e** Enhancer activity is decreased in anterior neural plate and secondary notochord in presence of mutation chr6:28950050 G > A. **c** is the wild-type enhancer for both **d** and **e**. **g** Increased expression in the epidermis is observed when the enhancer is mutated (chr6:28950885A>G)

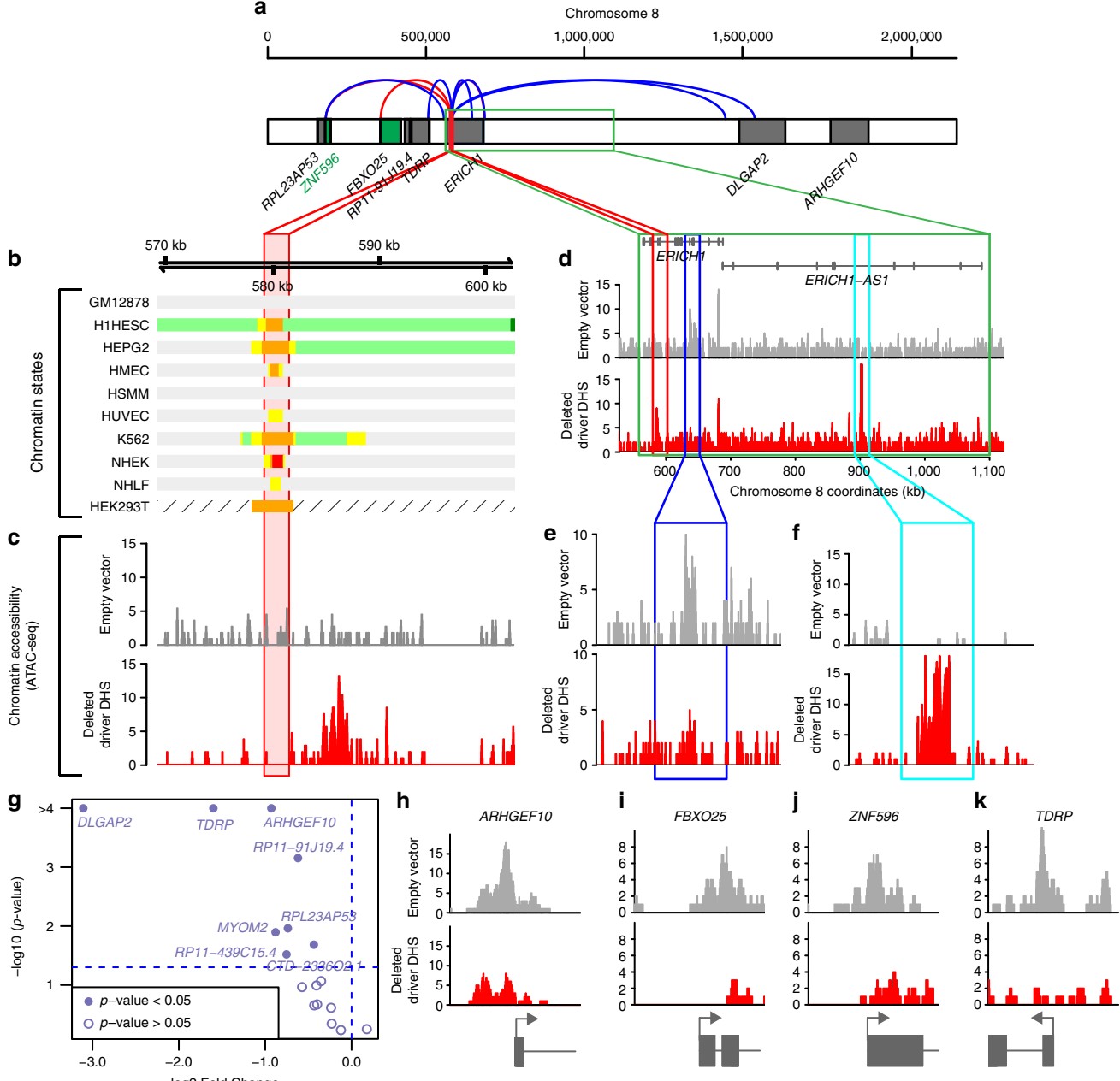

**Fig. 6** Deletion of putative driver DHS chr8:579137-581436 results in chromatin remodeling in the surrounding interval. **a** The relative positions of the deleted putative driver DHS (*red*) and the neighboring genes expressed in HEK293T are shown. ZNF596 (*green*) had altered expression associated with mutations and known interactions (Filter 3). *Curved lines* show validated (*blue*) and predicted interactions (*red*) between the driver DHS and putative target genes[25, 26]. **b** Chromatin states of HEK293T were derived from ChIP-seq data in the GEO series GSE51633[61] and for the remaining cell lines from the Broad ChromHMM track in the UCSC genome browser[70] (*light green* = weakly transcribed; *dark green* = transcribed, *orange* = strong enhancers, *yellow* = weak enhancers, *red* = active promoters, *light red* = weak promoters, *violet* = inactive/poised promoters, *blue* = distal CTCF/insulators, *dark gray* = polycomb repressed, *light gray* = heterochromatin, hatched box = missing data). **c** Show ATAC-seq data corresponding to deleted putative driver DHS (*red*) and HEK293T cell line treated with empty vector (*gray*). **d**–**f** The deletion of DHS chr8:579137-581436 (highlighted in *red*) results in the loss of an ATAC-seq peak (40 kb distal, *dark blue square*, shown in **e**) and the gain of an ATAC-seq peak (410 kb distal, *light blue square*, shown in **f**). **g** Volcano plot showing log2 fold change and *p*-value of gene expression differences (Wald test, computed using the DESeq function in R) between cells with deleted putative driver DHS and cells treated with empty vector. **h**–**k** Shown are ATAC-seq peaks in four genes that have alterations in chromatin accessibility at their promoters (indicated by *arrow*) when the putative driver DHS is deleted

DNA extraction using AllPrep DNA extraction kits (Qiagen GmbH, Hilden, Germany); DNA was extracted from blood using Qiagen Clotspin Baskets and DNA QIAmp DNA Blood maxi kits (Qiagen Inc., Valencia, CA, USA). DNA concentrations for all tumor and blood samples were determined by fluorometry (Qubit, Life Technologies). ER, PR and HER2 status for the 47 TCGA discovery samples and 50 TCGA replication samples were obtained from the TCGA Data

Portal (https://tcga-data.nci.nih.gov/tcga/). Clinically relevant breast cancer phenotypes were derived from ER, PR and HER2 statuses as follows[6]: (1) hormone receptor (HR) positive (+) includes tumors that are ER+ and/or PR+ and HER2 negative (−); (2) HR+/HER2+ includes tumors that are ER+ and/or PR+ and HER2 +; (3) HR-/HER2+ includes tumors that are ER− and PR− but HER2+; and 4) triple negative includes tumors that are ER−, PR− and HER2−.

**Genome analysis**. For the discovery and replication samples obtained from TCGA, Samtools 0.1.18[45] was used to remove duplicate reads from the BAM files. Base quality score recalibration was performed using GATK v1.6-5-g557da77[46] in two steps: first, the recalibration table was created using *CountCovariates*, then base quality scores were updated using *TableRecalibration*. Local realignment around indels was performed using GATK *RealignerTargetCreator* and *IndelRealigner* (Supplementary Data 12). MuTect[15] was used to call somatic point substitutions in the 97 genomes. Mutations were filtered to reduce the number of false positives; only somatic substitutions with at least 14× coverage in the tumor, 8× in the matched normal and with an allelic fraction ≥10% in the tumor were retained[15]. We performed two additional filtering steps. First, all somatic mutations overlapping with the 5,298,130 repetitive elements (as determined using *IntersectBed*[47] and the *RepeatMasker* track from UCSC Genome Browser)[48] were removed. Second, somatic mutations overlapping any of the 53,567,890 loci included in the dbSNP 137 database[49] were discarded (Supplementary Table 2). To examine coding somatic mutations, coordinates of all mutations were intersected with coding exons included in RefSeq (V55) (Figs. 1b, c) using *IntersectBed*.

**Kataegis analysis**. Rainfall plots were made as described by Nik-Zainal et al.[18]: briefly, for each mutation, the intermutation distance used to construct the plots was calculated as the mean distance to its two neighboring mutations (upstream and downstream) (Supplementary Table 3). All stretches of six or more consecutive mutations with a mean intermutation distance shorter than 1,000 bp were considered as kataegis loci[19, 50].

**Identification of significantly mutated breast DHSs in Filter 1**. A total of 266,757 DHSs from a HMF line[51] and 278,680 DHSs from the T-47D breast cancer cell line[52] were obtained from the UCSC Genome Browser composite track *wgEncodeUwDgf*[22]. After removing DHSs overlapping known RefSeq genes there were 237,046 HMF DHSs and 247,051 T-47D DHSs. Overlapping DHSs (by at least one base pair) were combined using BEDTools v2.20.1 *MergeBed*[47] resulting in 392,977 unique DHSs (181.6 Mbp). Fifty-five DHSs were removed from the analysis because they overlap kataegis loci (Supplementary Table 3). Repetitive elements, derived from the *RepeatMasker* track of the UCSC Genome Browser[48], on DHSs were masked, reducing the corresponding DHS length. The 58,141 DHSs with remaining length zero after repeat masking were omitted from the analysis. The 334,781 DHSs that do not overlap kataegis loci and have remaining length larger than zero were then intersected with the mutations in the 47 discovery breast cancer samples using BEDTools *intersectBed*.

**Clustering DHSs into groups of similar sequence characteristics**. The signature of positive selection for mutations at a driver locus in the tumor genome is an increase in observed mutation density relative to the expected density under a neutral model of evolution. We developed a statistical test to detect this kind of increase in local mutation density on DNase hypersensitivity sites. This test relies on an inference of the local mutation probability of DNA sequences, which is known to vary considerably across the tumor genome. Several sequence characteristics, such as gene expression levels and GC content, are known to influence the local mutation density[53]; therefore comparing the mutation probabilities of DHSs with similar sequence characteristics enables the identification of those that are mutated at a rate higher than expected by chance[15]. To control for all factors affecting mutation probability outside of selection, we took five covariates into account by performing a *k*-means clustering procedure on all DHSs per chromosome: (1) DNA replication timing[23], (2) open and closed chromatin status measured by HiC mapping (GEO series GSE35156)[31], (3) GC content[53], (4) local gene density[15], and (5) expected mutations on the basis of trinucleotide content. Local gene density for each DHS was calculated as the fraction of base pairs within 250 kbp overlapping a RefSeq gene (BEDTools *WindowBed –w 250000*[47] was used to extract the genes associated with each DHS), while the other values were retrieved directly from the indicated references. To determine the expected context-dependent mutation probability up to a constant factor, the genomic trinucleotide distribution and the number of mutations per trinucleotide were calculated. For each trinucleotide *t*, we counted the number of mutations $M_t$ and the number of its occurrences $N_t$ throughout the genome (centromeric regions excluded). For each DHS *i*, we then calculated the expected mutation rate $\mu_i$ as:

$$\mu_i = \sum_{t=\text{AAA}}^{\text{TTT}} \frac{n_{i,t} M_t}{N_t} \qquad (1)$$

where $n_{i,t}$ is the number of occurrences of trinucleotide *t* in the DHS *i*. A value for each covariate was calculated for every DHS, and all covariates were normalized to have mean = 0 and standard deviation = 1. Breast DHSs were binned into 223 clusters. Each cluster included at least 100 DHSs, for a total length of at least 20 kb (Supplementary Data 4).

**Generation of DHS clusters from control tissues**. The coordinates of DHSs active in 53 cell lines were downloaded from the UCSC Genome Browser composite track *wgEncodeUwDgf*[22]. DHSs associated with the same tissue type were merged using BEDTools *mergeBed*, resulting in 13 distinct tissues (Supplementary

Table 4). DHSs in control tissues that overlapped with any of the 392,977 breast DHSs were removed from the control set for the significance test (Supplementary Table 5). Thus, the set of breast DHSs is more enriched in constitutive DHSs (Supplementary Fig. 7a), i.e. those that are active in many different tissue types. Control tissues DHSs were clustered into groups to account for the five covariates of local mutation rate as described above for the breast DHSs. In this manner, we generated 13 independent sets of control tissues DHSs.

**Inference of mutation probability**. Clustering of DHSs according to five covariates of mutation rate was assumed to account for all relevant factors affecting local mutation rate. Consequently, the per-nucleotide neutral mutation probability $\mu$ within a given cluster was assumed constant across all of its member DHSs ($i = 1, …, N$). This left two remaining factors that would have an effect on the observed number of mutations *n* on a given DHS from the cluster: 1) the length *L* of the DHS (after repeat masking), which determines the mutational target size, and 2) selection. For a DHS sequence that evolves neutrally we expect $n \sim \text{Poiss}(\lambda)$, where $\lambda = \mu L$, while positive selection increases $\lambda$ through its effect on the fixation probability of a mutation, i.e., an increase in $\mu$. We inferred the mutation probability per cluster using the maximum likelihood estimator:

$$\hat{\mu} = \frac{\sum_{i=1}^{N} n_i}{\sum_{i=1}^{N} L_i} \qquad (2)$$

Note that this value will be upward-biased from the true neutral probability if a cluster contains DHSs under selection. Supplementary Fig. 8a shows an illustrative example of a representative distribution of mutation counts, $P(n)$, from a cluster of DHSs. The cluster is constructed both from sites that evolve neutrally as well as sites under positive selection. Inference according to Eq. 2 in this case entails an overestimation of the true neutral mutation probability that governs the evolution of neutral sites. Importantly, this means that our test statistic, the probability *p* of observing as many or more mutations on a DHS, is a conservative measure and by construction overestimates the true *p* of a DHS under selection.

**Test statistic p**. Given the estimated mutation probability of a DHS from a given cluster inferred as in Eq. 2, we computed the probability of observing at least *n* mutations in the DHS of length *L* by:

$$p = 1 - \sum_{k=0}^{n-1} \text{Poiss}(k; \hat{\lambda}) = 1 - \sum_{k=0}^{n-1} \frac{\hat{\lambda}^k}{k!} e^{-\hat{\lambda}} \qquad (3)$$

with $\hat{\lambda} = \hat{\mu} L$. We used *p* defined in Eq. 3 as *p*-value to measure the deviation of mutation counts on a DHS from the null expectation of neutral evolution. Finally, we combined the data from all clusters of DHSs to obtain a distribution of values *p* of DHSs from all chromosomes and all mutation rate backgrounds. This distribution will be considered in the following sections and was used to test for the presence of significantly mutated DHSs.

The observed values *p* from cancer data need to be compared to the expected values, whose distribution is however not known a priori, due to the Poisson nature of the mutation count *n*. We simulated the expected distribution of *p* from 100 random instances of $n_i \sim \text{Poiss}(\hat{\mu}_i L_i)$ per DHS *i*, with $\hat{\mu}_i$ inferred at the cluster level as described above (Eq. 2). This yielded a null model of neutral evolution for both breast and control tissues and therefore also quantified any remnant deviation in control tissue DHSs from the neutral expectation.

Supplementary Fig. 8b shows the normalized histogram of values *p* obtained from the cluster distribution $P(n)$ shown in Supplementary Fig. 8a. Because $n \sim \text{Poiss}(\lambda)$, *p*-values are non-uniformly distributed, with a pronounced peak at $p = 1$ for the relevant range of small to intermediate $\lambda$. Overestimating the true $\mu$ through $\hat{\mu}$ entails that the values *p* of the selected sites in the cluster are overestimated as well, as their observed mutation counts *n* are less extreme than they would be with the true and smaller $\lambda = \mu L$. Supplementary Fig. 8b illustrates why, therefore, the inferred *p*-value *p* is a conservative estimate of the true *p*-value of a DHS under positive selection. Furthermore, the observed distribution of *p* has a larger variance than the expected distribution, owing to its composition (neutral and selected components) of different mutation probabilities. Therefore, in the depicted case of positive selection for driver mutations, the random simulated distribution of *p* will have a higher proportion of sites with $p < 1$ than the observed distribution (Supplementary Fig. 8b). This is in line with the excess of the expected *p* for breast DHSs at intermediate values $0.1 < p < 1$ shown in Supplementary Fig. 2. As expected from the assumption of neutral evolution, this trend is greatly reduced for the control tissues (Supplementary Fig. 9).

**False discovery rate**. The expected distribution of *p* under the model of neutral evolution allows for an estimate of the fraction of values *p* among the observed

ones that occur by chance, as a function of a *p*-value threshold *p\**:

$$\mathrm{FDR}(p*) = \frac{F_{\exp}(p*)}{F_{\mathrm{obs}}(p*)} \qquad (4)$$

where $F(p)$ denotes the cumulative distribution function at *p*. Figure 2c shows the FDR of Eq. 4 for breast as well as control tissue DHSs. We set the allowed fraction of false rejections of the null hypothesis of neutral evolution to FDR($p*$) = 0.25, which defines a threshold $p*$ for each tissue below which DHSs were considered significant.

**Testing the Poisson assumption.** Our statistical test relies on the assumption that mutation counts on DHSs are Poisson random variables. We tested this assumption using the set of DHSs that are active in control tissues, serving as a proxy for neutrally evolving sites. Given the large number of DHSs and correspondingly smooth distribution of *p* in each tissue, we used the Kolmogorov–Smirnov test to assess the probability with which the observed values *p* in control tissues stem from the expected distribution simulated under the neutral Poisson model. Conversely, if there exist breast DHSs under selection, we would expect a low similarity. Supplementary Table 5 shows the *p*-value of the KS test statistic, with low values indicating that the two data sets, observed and simulated neutral, are unlikely to have been generated from the same underlying distribution. We found that all control tissues exhibit almost perfect agreement with the random expectation, while breast DHSs have a distribution of *p* that cannot be matched, indicating selection. This does not imply that mutations on breast DHSs under positive selection are not Poisson distributed, but rather that their Poisson distribution parameter $\lambda$ is larger than that of neutrally evolving sites. This is because positive selection increases the probability of fixation of each positively selected mutation arising on the sequence. From this we conclude that the Poisson assumption of mutation counts is in good agreement with the observed data.

**Inferred mutation probability and DHS ubiquity dependence.** Supplementary Fig. 7b shows that inferred mutation probabilities $\hat{\mu}$ (Eq. 2) in clusters of breast DHSs are on average elevated compared to those on non-breast DHSs. DHSs in control tissues that overlapped with any of the 392,977 breast DHSs were removed from the control set for the significance test (Supplementary Table 5). Thus, the construction of the breast and control tissue data sets (see above), resulted in breast DHSs being enriched in constitutive DHSs. Here we examined whether these observed differences affect the inference of significantly mutated DHSs in breast vs. control tissues. To test whether the excess of mutations in our set of significantly mutated breast DHSs derives from a biased mutation probability in the subset of clusters the DHSs originate from, we compared the distribution of inferred mutation probabilities across all clusters to that from the subset of DHSs with $p < p*$ Supplementary Fig. 10a shows that the two distributions are very similar, suggesting that the excess of mutations cannot be attributed to a difference in the background mutational properties of the subset of highly mutated DHSs. As it is observed, we expected a slight upward shift in mean $\hat{\mu}$ among the ensemble of significant DHSs, since the clusters which the selected sites originate from, by construction, have higher inferred mutation probabilities than their fully neutral counterparts. With respect to tissue specificity, we found that the highly significant DHSs are slightly overrepresented among the very ubiquitous ones (active in 13 and 14 tissues; see Supplementary Fig. 10b), but largely cover the whole range of tissue specificity. From this we conclude that tissue specificity does not have a large effect on our inference of selection.

**Quality assessment of significantly mutated DHSs in Filter 2.** Three measures of sequencing and alignment quality were assessed for each DHS using BAM files from 94 samples (47 tumors and their matched 47 germline DNA): (1) mean read count over each position in the DHS; (2) mean read quality; and (3) fraction of improperly aligned reads, calculated as the number of reads that have abnormal pairing in terms of orientation and distance divided by the total number of read pairs aligned on the DHS. For each cluster of DHSs, mean and standard deviation were calculated for all three quality measurements and *Z*-scores were calculated for each DHS as the difference between its value and the cluster mean, divided by the cluster's standard deviation. DHSs were eliminated if at least one of three conditions occurred: (1) read count *Z*-score lower than −2; (2) read quality *Z*-score < −2; and (3) improperly aligned reads *Z*-score > 2.

**Evidence for interactions between DHSs and promoters in Filter 3.** Two independent experimental data sets were used to identify putative target genes of the 14,087 mutated DHSs (Fig. 2b): (1) 3,095,882 interactions including 3C, 4C, 5C, ChIA-PET, Hi-C, Capture-C, and IM-PET included in 4DGenome[26] (February 1st 2015); and (2) A collection of 1,454,901 predicted distal regulatory and promoter interactions derived from the correlation of DHS peaks in promoters and distal elements from 79 cell types in the ENCODE study[25]. The coordinates of all the interactions from both sources were intersected with the 14,087 mutated DHSs using BEDTools *IntersectBed*. If the coordinates of one end of the interaction overlapped with a mutated DHS and the coordinates of the other end intersected with a promoter or gene body, the gene was marked as a validated target of the mutated DHS.

**Gene expression analysis.** RNA-seq data normalized and reporting the expression levels of 20,531 genes for the 47 discovery breast cancers and 106 unrelated normal breast tissues was downloaded from TCGA (https://tcga-data.nci.nih.gov/tcga/dataAccessMatrix.htm). CNV data for all the 47 discovery breast cancers was also downloaded from TCGA. Only genes not overlapping CNVs in the sample where the DHS was mutated were considered for the analysis. Expression levels of each gene associated with one or more mutated DHS were normalized across the panel of 106 normal breast tissues using the calcNormFactors function from the EdgeR package in R[54]. For each gene, the normalized expression level was compared to the expression level in breast cancer samples with an associated mutated DHS using the Wilcoxon test. Genes showing significantly different expression levels (*p*-value < 0.05) were considered aberrantly expressed and a putative target of the associated mutated DHS(s).

**Replication screenings in Filter 4.** Two relatively small breast cancer datasets were jointly used in replication set 1 (50 tumors with WGS from TCGA and 135 tumors that underwent targeted sequencing, see below) and one large breast cancer dataset was used in replication set 2 (560 breast cancers with WGS derived from the BRCA-EU project[2]). For replication set 2, the list of 3,851,143 mutations was downloaded from ICGC Release 22 (frozen at August 24, 2016, https://dcc.icgc.org/releases).

**Targeted sequencing of selected regions.** Deep-targeted sequencing of 135 matched tumor and blood samples was performed using the Illumina TruSeq Custom Amplicon kit. Using DesignStudio software, probes were successfully designed to cover 46 of the 73 putative driver DHSs and 94.0% (171 of 182) coding exons of the 12 genes most highly mutated in breast cancer[8] (Supplementary Data 13). Sequencing libraries were prepared following the TruSeq Custom Amplicon Library Preparation Guide with the following modifications[55]. For all samples, 500 ng of DNA was used. Extension-ligation was performed according to manufacturer's protocol. Two PCR cycles were performed as follows: 95 °C 3 min, followed by 2 cycles of 95 °C for 30 s, 66 °C for 30 s, and 72 °C for 60 s. Following two cycles of polymerase chain reaction (PCR), Ampure Bead cleanup was carried out according to standard protocol and eluted in 25 µl. Following cleanup, 20 µl of PCR product, 22 µl of PMM2/TDP1 and 4 µl of the i7 primer supplied with from the TruSeq Custom Amplicon Index Kit were added to PCR tubes, and PCR was carried out using the following conditions: 95 °C for 30 s; 22 cycles of 95 °C for 30 s, 66 °C for 30 s, and 72 °C for 60 s; and finally 72 °C for 5 min (note that number of cycles is specific to TruSeq Custom Amplicon Design). Ampure bead cleanup was performed, library quality was assessed on an Agilent Bioanalyzer (Agilent Santa Clara, CA, USA) using a DNA 1000 chip, and concentration determined by Qubit and quantitative PCR using the KAPA Library Quantification Kit (Kapa Biosystems, Woburn, MA, USA). Samples were pooled and 45 samples per run were sequenced at 12 pM using Illumina MiSeq V2 sequencing reagents and the following run set up: Read 1: 301 cycles; Read 2: 8 cycles; Read 3: 8 cycles; Read 4: 301 cycles.

**Targeted sequencing read processing and variant calling.** FASTQ files were retrieved from Illumina Basespace (https://basespace.illumina.com/). Mutascope was used to process reads and to prepare them for variant calling[56] because it is a high-sensitivity software suite designed to analyze data derived from high-throughput sequencing of PCR amplicons from matched tumor/normal samples. Briefly, Mutascope requires FASTQ files and the list of coordinates of the amplicons as input. It first creates a blacklist of mutations that might be identified because of misalignments of reads or alignment to homologous regions in the genome. Reads are then aligned using BWA-Smith Waterman alignment algorithm[46]. Low-quality reads are removed and primer sequences are soft-clipped. Read groups are created to associate each read to a specific amplicon. GATK is subsequently used to realign reads around indels in both the tumor and normal samples. Reads with a high Smith-Waterman score and mapped at the expected location were used for further analysis (Supplementary Fig. 11).

Germline variants were called using Mutascope and *Hclust* in R was used to cluster samples on the basis of their genotypes at 1,020 dbSNP loci to ensure that matched tumor and blood samples were correctly paired. Mutations were identified using both Mutascope variant caller[56] and MuTect[15]. As Mutascope is optimized for variant calling in PCR amplicons that have fixed start and end positions, it estimates the error rate at each genomic position based on the nucleotide, position in the read and read type (i.e., forward or reverse), and compares the allelic fraction of each mutation with the error rate using a binomial test for significance. Mutascope discriminates between somatic mutations and germline variants using a Fisher's exact test and filters false positives based on biases in read groups (mutations called only in forward or reverse reads) or low quality of base calls. Mutations were independently called by Mutascope and MuTect (as described above) then the union of the results from the two methods was retained. Mutations were further filtered to eliminate those with a tumor allele frequency <5% or normal allele frequency >5%.

**Mutation analysis in 19 tumor types**. Metadata associated with 1,097 tumors belonging to 19 different cancer types (with no use restrictions) in TCGA was downloaded from the Cancer Genomics Hub (https://browser.cghub.ucsc.edu/) (September 30, 2014). GTFuse (http://fuse.sourceforge.net/) provides remote access of BAM files stored at the Cancer Genomics Hub. The 1,097 tumor and matched germline BAM files were accessed and sequence data was extracted using *samtools view* corresponding to the 10 putative driver DHSs (plus 50kbp upstream and downstream of each element) and, to serve as positive controls the exons of 12 cancer genes (Supplementary Fig. 5) (*CTCF, CTNNB1, EGFR, FLT3, IDH1, IDH2, NFE2L2, NRAS, PIK3R1, PTEN, TP53* and *VHL*) identified as high-confidence drivers by four independent methods in the TCGA PanCancer analysis[27]. Variants were called with MuTect (using the filters described above), i.e., all mutations labeled as "KEEP" by MuTect, with at least 14× coverage in the tumor and 8× in the normal, and at least 10% allele frequency in the tumor were retained. All mutations in repeat elements were discarded.

To determine if the 16 putative driver DHSs were enriched for mutations across all 19 tumor types we performed two analyses. First, we performed an interval analysis in which the mutation density for each of the 16 putative driver DHSs was calculated as the ratio between the number of mutations in all tumor types and the number of non-repetitive nucleotides. Mutation probabilities were then calculated for the intervals (+/−50 kb) surrounding each putative driver DHS. Poisson $\lambda$ parameter was estimated from the mutation probability in the interval surrounding each DHS and *p*-values *p* were calculated as shown in Eq. 3. We also performed a cluster analysis in which the mutation density for each of the 16 putative driver DHSs was compared with the mutation density of DHSs with similar sequence characteristics as described above for the breast cancer whole-genome sequence analysis (Supplementary Data 4). For each of the 16 putative driver DHSs, 20 random DHSs were selected from the corresponding cluster and GTFuse was used to extract these sequences from the 1,079 tumors and matched germline BAM files. Mutations were identified using MuTect and values *p* were calculated as shown in Eq. 3 to identify significant mutation density differences between each of the 16 putative driver DHSs and their corresponding randomly selected 20 DHSs. The significance threshold in both tests was set to FDR <0.05, corresponding to $p < 0.017$ and $p < 0.013$, respectively (Supplementary Table 8, Supplementary Data 10).

**In silico analysis of DHS chr5:1325957-1328153**. The putative driver DHS chr5:1325957-1328153 located upstream of *TERT* is mutated in seven breast cancer TCGA samples (four discovery, three replication). We analyzed the expression levels of the 15 genes within 500 kb distance using the same method as described in Filter 3: normalized expression levels (RNA-seq) were downloaded from TCGA and compared with 106 unrelated normal breast samples using Wilcoxon test. Genes showing significantly different expression levels (*p*-value < 0.05) were considered aberrantly expressed.

For the focused mutation analysis of the *TERT* promoter (located in the 50 kb region upstream of the putative driver DHS on chr5:1325957-1328153), the filter on repetitive elements was removed because the two reported positions mutated at high frequency[33] are in a repeat element.

**In vivo effects of mutations in DHS chr6:28948439-28951450**. *C. intestinalis* were obtained from M-Rep (San Diego, CA, USA). Eggs were fertilized in vitro and electroporated with wild-type or putative mutant DHSs reporter constructs as described in Christiaen et al.[57] Primers used to build constructs are shown in Supplementary Table 11. Embryos were then fixed at 8 h post fertilization for 15 min in 4% formaldehyde and prepared for microscopy as described in Farley et al.[35]. After mounting embryos on slides, slide labels were covered and randomized for blind analysis. At least 45 embryos were analyzed on each slide and at least two biological replicates were performed for each mutation. Replicates were combined and differences in GFP expression between wild-type and mutated DHS were assessed using Fisher's exact test.

**Deletion of driver DHSs in HEK293T cell line**. The HEK293T cell line[58] was chosen based on: (1) transfection has very high efficiency[59]; (2) analysis of genes using BioGPS[60] within 500 kbp of the deleted putative driver DHSs showed that they are expressed in the cell line; (3) analysis of ChIP-seq data from HEK293T cells (Fig. 6b, Supplementary Fig. 6) showed that the chromatin states of the analyzed putative driver DHSs and their associated genes are similar to those derived from ChromHMM for nine different tissues types. BigWig files of ChIP-seq data from HEK293T cells were retrieved from the Gene Expression Omnibus (GEO) series GSE51633[61], sample IDs GSM1249885 (H3K4me3), GSM1249886 (H3K4me2), GSM1249887 (H3K4me1), GSM1249888 (H3K4me1) and GSM1249889 (H3K27ac) and visualized using the UCSC genome browser.

**Short guided RNA design and construction**. Short guided RNA (sgRNA) targeting 5′ and 3′ sequences of the putative driver DHS (target regions) and control region were designed and analyzed for specificity using CRISPR Design Tool—7-21-2013[62] (Supplementary Fig. 12a, b). The sgRNA expression plasmids were prepared by cloning annealed oligos into the PHI74 vector digested by BbsI. sgRNA oligos are shown in Supplementary Table 12.

**Selection of homozygous clones**. HEK293T cells were cultured in Dulbecco's modified Eagle medium (Life Technologies) containing 10% fetal bovine serum (Life Technologies), 2 mM glutamine (Life Technologies), 7 mM maximal exact matches (MEM) non-essential amino acids (Life Technologies), 1 mM sodium pyruvate (Life Technologies), 10 mM HEPES,100 units/ml penicillin streptomycin, and 100 μg/ml streptomycin (Life Technologies). Cells were co-transfected with constructs expressing 5′ and 3′ sgRNAs and pBABE-Puro[63] in Opti-MEM reduced serum medium (Life Technologies) using the Lipofectamine 3000 Reagent (Life Technologies). Cells were co-transfected in 48-well plates with either 0.75 μg of 5′sgRNA, 0.75 μg, 3′ sgRNA and 0.5 μg pBABE-Puro or 1.5 μg of empty PHI74 vector and 0.5 μg pBABE-Puro. Forty-eight hours after transfection, HEK293T cells were cultured for 14 days in the presence of Puromycin (0.75 μg/ml). 72 h after transfection cells were split onto 10 cm dishes to ensure clonal growth. 10–14 days after splitting, clones were picked, amplified and tested for CRISPR-CaS11-mediated deletion by PCR. Homozygous clones were further amplified. Clones were harvested on different dates and pellets were frozen; however, all samples were processed simultaneously for DNA and RNA extraction (ATAC-seq and RNA-seq), library generation and sequencing.

**PCR strategy for detection of CRISPR-Cas9-mediated deletion**. PCR was performed using Phusion Hot Start II High-Fidelity DNA Polymerase (Thermo Scientific). PCR strategy for detection of CRIPR-Cas9-mediated deletion is depicted in Supplementary Fig. 12c, d. Primers are listed in Supplementary Table 13. PCR was performed in a total volume of 25 μl with 50 pM of each primer, 400 μM dNTPs (total), 0.5U Polymerase and 10 ng of genomic DNA.

**ATAC-Seq data processing**. Nuclei were isolated from 50,000 cells and incubated with 2.2 μl Tn5 Transposase (Illumina) for 30 min at 37 °C[64]. DNA was immediately purified using Qiagen MinElute columns. Library amplification required ten cycles and was performed using KAPA Biosystems Real-Time Library Amplification Kit. Libraries were size selected to between 200 and 800 bp using the SPRIselect Reagent Kit (Beckman Coulter) and sequenced PE1 using an Illumina HiSeq2500. FASTQ files were retrieved from Illumina Basespace (https://basespace.illumina.com/) and ATAC-seq reads were aligned to the human reference genome (hg19) using BWA with MEM[46]. Paired-end reads were separated to perform alignments and then were merged using Picard Tools v. 1.115 MergeSamFiles (http://picard.sourceforge.net). Duplicate reads were marked with Picard Tools MarkDuplicates (Supplementary Fig. 13a), read depth at each position in the surrounding 1 Mb genomic interval (+/− 500 kb) was determined using Bed Tools v. 2.20.1 genomeCoverageBed[65]. We normalized the read depth at each genomic position by considering the total number of mapped reads in the analyzed sample compared to all samples (Fig. 6).

**RNA-Seq data processing**. Total RNA was assessed for quality using an Agilent Tapestation, and samples determined to have an RNA integrity number of 7 or greater were used to generate RNA libraries using Illumina's TruSeq Stranded Total RNA Sample Prep Kit. RNA libraries were multiplexed and sequenced with 125 bp paired end reads (PE100) to a depth of approximately 25 million reads per sample on an Illumina HiSeq2500. RNA-seq reads were aligned to the human genome (hg19) with STAR 2.4.0 h (outFilterMultimapNmax 20, outFilterMismatchNmax 999, outFilterMismatchNoverLmax 0.04, outFilterIntronMotifs RemoveNoncanonicalUnannotated, outSJfilterOverhangMin 6 6 6 6, seedSearchStartLmax 20, alignSJDBoverhangMin 1) using a splice junction database constructed from Gencode v19[66, 67]. Reads overlapping genes were counted using HTSeq-count (-s reverse -a 0 -t exon -i gene_id -m union) (Supplementary Table 14, Supplementary Data 14, Supplementary Fig. 13b)[68]. Raw read counts were processed with DESeq2[69] and only genes with mean read count >20 were considered for the analysis. Technical replicates were collapsed using DESeq2 collapseReplicates. Read counts were transformed using variance stabilizing transformation[68]. Differential expression analysis of the nine genes in the region of interest (i.e. within the 1Mbp interval surrounding the putative driver DHS at chr8:579137-581436) was performed using the DESeq2 function with default parameters[69].

**Data availability**. Data is available at ArrayExpress: E-MTAB-5710 (targeted sequencing); E-MTAB-5714 (RNA-seq); E-MTAB-5702 (ATAC-seq). All other remaining data are available within the Article and Supplementary Files, or available from the authors upon request.

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

## Acknowledgements

This work was supported by NIH grants UL1TR000100 and P30CA023100. Equipment in the Neuroscience Microscopy core was used for the Ciona experiments; this core is supported by the UCSD School of Medicine Microscopy Core Grant P30 NS047101. CD is supported in part by the University of California, San Diego, Genetics Training Program through an institutional training grant from the National Institute of General Medical Sciences (T32GM008666) and the California Institute for Regenerative Medicine (CIRM) Interdisciplinary Stem Cell Training Program at UCSD II (TG2-01154). HA is supported by a grant from the German Cancer Aid (#109790). We thank Kristen Jepsen, Hiroko Matsui, Aquila Fatima, Mahdieh Khosroheidari and Marika Smith for assistance. We thank Sharmeela Kaushal and Scott Vandenberg for providing breast cancer samples. We thank Richard Kolodner, Erin Smith and Olivier Harismendy for comments.

## Author contributions

K.A.F. and M.D. conceived the study. M.D., D.W. and C.D. performed data processing and computational analyses. D.W. and S.R.S. conceived the statistical analysis for Filter 1. K.M.O. and E.K.F. performed *C. intestinalis* experiments and analyzed the data. A.L.R. and R.B.S. provided tumor and matched germline samples. A.D.C., A.A. and H.A. prepared samples for targeted sequencing. R.B.S. provided insight and support for the design and execution of the project. M.D. and K.A.F. prepared the manuscript.

## Additional information

**Competing interests:** The authors declare no competing financial interests.

