## [Peer Review File · Nature Communications]

Reviewers' comments:

Reviewer #1 (Remarks to the Author):

The authors undertake an important and ambitious analysis. They attempt to develop a systematic approach to identify distal regulatory elements in breast cancer. The authors demonstrate that the 47 breast cancer samples selected are a reasonable representation of breast cancer in general. They identify a number of candidates and explore one of the candidates near TERT and show that it likely regulates the levels of a number of transcripts from genes in the region. However, it is important to emphasize that the studies in this manuscript are solely associations without any interventions to validate whether the putative driver distal regulatory elements are indeed drivers.

The manuscript has a number of interesting points but also concerns.

1. The authors analysed 47 breast cancers and used 50 as a replication set. There are other datasets with whole genome analysis. These should be considered for analysis in this study.
2. While combining breast cancer samples to gain power in a small sample set does make sense, there is no more reason to combine basal breast cancer with luminal breast cancer than there is to use all whole genome sequences across the complete TCGA data. Why did the authors not use the complete TCGA set for discovery or a split of the data set for discovery and testing. Indeed, this would strengthen the manuscript as there are over 1000 samples with whole genome sequencing. This would complement the current analysis.
3. The authors test the potential distal regulatory elements across a separate breast cancer set. What is the distribution of the specific distal regulatory elements across different breast cancer subtypes. One would expect selectivity. This should be noted in the text.
4. The authors do explore one distal regulatory element near TERT in further detail. This is a positive. However, given that as they indicate in the introduction, TERT is the only gene where promoter mutations have been validated to be important in cancer, more detailed exploration of additional sites particularly the one that is present in most tumor lineages would strengthen the manuscript.
5. The authors at all stages took approaches that would decrease the false negative rate and would increase FDR. For example an FDR of 125 is very liberal. This is OK but should be noted in the discussion of the data. It also warrants care in the text. "Using this threshold we identified 637 significantly mutated breast DHSs". This statement on page 4 of the text should have a conditional. These should be listed at this point as potentially significantly mutated breast DHSs. They are not yet significantly mutated.
6. The authors use "control" tissues to test the potential that the DHSs identified in breast cancer have undergone positive selection. This is a good addition to the manuscript. It would be strengthened with more samples than the 13 tissues used. Further the authors claim Indeed, we observe a significantly lower signal than in breast DHSs (Fig. 2C), albeit still positive, suggesting that there are DHSs active in the control tissues that may be positively selected for mutations in breast cancer. As indicated above, there is a second possibility which is that with a FDR of .25, there may be false positives in the normal tissue. The authors should consider or at least comment on this possibility. The subsequent arguments are reasonable.
7. The enrichment of potentially selected DHS at the different filter levels argues for enrichment and is a strong positive. However, it does not speak to any of the specific DHSs that are being explored. This should be noted. This is picked up in the additional filters further along.
8. The gene ontology analysis is done on an unvalidated set of DHS. This should not be done at this stage but rather after Filter 4 is performed. The associations with the 16 putative driver DHSs would be stronger.
9. Overexpression of cancer associated genes. This is potentially the most important part of this

study. It is important to emphasize that all of the studies presented are associations and not evidence of causality. One question that is not approached in this section is whether the putative driver DHS are mutually exclusive with promoter mutations for hTERT. It may be that there is not sufficient power for this comparison..

10. A key question would be to determine whether introduction of the specific changes in the DHS and preferably ones that are not near hTERT actually increase the transcription of genes of interest. This was a requirement for the demonstration that the TERT promoter mutations are indeed drivers of hTERT expression.

Reviewer #2 (Remarks to the Author):

This paper describes a strategy for identifying candidate driver mutations in breast cancers using whole genome sequences from tumor samples. The main features are to (a) focus on candidate regulatory regions using chromatin accessibility data (DNase hypersensitive sites or DHSs) measured in relevant cell types and (b) identify DHSs with mutation rates in the tumors that are significantly higher than expected under a neutral mutation model. The authors apply several filters to focus on DHSs containing mutations more likely to be drivers of tumors. One of the three DHSs containing likely driver mutations was shown to be associated with genes with altered levels of expression in tumors, leading to the hypothesis that the candidate driver mutations in this one DHS are affecting expression levels of several genes, including the gene TERT, which has been previously implicated in tumors.

The novel part of this report is developing a method for dealing with the five known genomic and epigenomic features that affect the local rate of mutations, specifically "1) DNA replication timing, 2) open and closed chromatin status, 3) GC content, 4) local gene density, and 5) expected mutations based on trinucleotide composition". The authors annotated almost 335,000 breast DHSs by these five features, and used k-means clustering to identify 223 clusters of DHSs similar in these features, with more than 100 DHSs per cluster. They inferred the background mutation probability for each cluster and identified DHSs that were mutated in breast cancer at a rate significantly higher than expected under a neutral model. The method developed to do this is a notable advance because it takes into account the several features that impact local rates of mutation, and it does so in a data-driven manner. The authors make a strong case that their estimates of enrichment and significance are conservative. Appropriate statistical tests are applied.

Identifying strong candidates for mutations or variants in regulatory regions that affect phenotypes is currently a topic of high interest. Many investigators are taking approaches largely examining epigenetic features such as histone modifications or transcription factor occupancy. While those approaches have high value, the current report shows the continuing value of rigorous searches for evidence of positive selection as a complementary guide for finding functionally significant mutations.

The report could be improved by addressing the following issues.

(1) The authors open the section on finding DHSs significantly mutated in breast cancer (page 4) with the comment that "the local mutation density is lower in regulatory elements that are active in the cell-type of origin of the tumor compared to regulatory elements in other cell types". Thus it may be puzzling to readers that the focus is on finding DHSs with a higher frequency of somatic mutations as an indicator of drivers. The answer is given in the Methods: "The signature of positive selection for mutations at a driver locus in the tumor genome is an increase in observed mutation density relative to the expected density under a neutral model of evolution." That explanation should also be stated clearly in the Results.

(2) After identifying 16 putative driver DHSs associated with breast cancer (top of page 6), the authors then focus on three DHSs with elevated mutation rates in multiple cancers. This leaves the impression that the other 13 putative breast cancer driver DHSs are not as important. The authors should clarify how readers should interpret these different sets of putative driver DHSs.

(3) While the authors focus on the putative driver DHS in an intron of the gene CLPTM1L (Figure 4, the other two pan-cancer driver DHSs are not discussed. One of them, chr8:579137-581436, is located close to a telomere. Other studies have shown a high rate of substitution close to telomeres (e.g. doi: 10.1073/pnas.1221792110). Could this chromosomal location contribute to the observed elevated mutation rate? The third DHS, chr20:62115827-62119284, does not appear to be a strong DHS. More discussion of these exemplar candidate driver DHSs, or alternatively, other breast cancer candidate drivers, would help readers appreciate the value and perhaps limitations of the results.

(4) page 4, top line, comments twice on the gene MAP3K1, but seemingly in two distinct groups; this is confusing.

(5) Fig. 2A: Filter 3 recovers 73 DHSs from an initial 596, but only 7 are shown as being removed. Is this an error? If not, where did the other DHSs go?

(6) Fig. 3A and 3B have no labels on the axes.

(7) Fig. 3C would be easier to understand if the description in the Results used the same phrases as the labels on the axes, i.e. clusters analyses and interval analyses.

Reviewer: Ross Hardison

Reviewer #1 (Remarks to the Author):

The authors undertake an important and ambitious analysis. They attempt to develop a systematic approach to identify distal regulatory elements in breast cancer. The authors demonstrate that the 47 breast cancer samples selected are a reasonable representation of breast cancer in general. They identify a number of candidates and explore one of the candidates near TERT and show that it likely regulates the levels of a number of transcripts from genes in the region. However, it is important to emphasize that the studies in this manuscript are solely associations without any interventions to validate whether the putative driver distal regulatory elements are indeed drivers.

The manuscript has a number of interesting points but also concerns.

1. The authors analyzed 47 breast cancers and used 50 as a replication set. There are other datasets with whole genome analysis. These should be considered for analysis in this study.

While this manuscript was under review, one additional WGS breast cancer dataset with 560 samples (BRCA-EU) was published¹. We agree with the reviewer that using these samples as a second replication set would improve the strength of our analysis.

[UNPUBLISHED DATA REDACTED BY EDITORIAL TEAM AS PER AUTHOR REQUEST]

2. While combining breast cancer samples to gain power in a small sample set does make sense, there is no more reason to combine basal breast cancer with luminal breast cancer than there is to use all whole genome sequences across the complete TCGA data. Why did the authors not use the complete TCGA set for discovery or a split of the data set for discovery and testing. Indeed, this would strengthen the manuscript as there are over 1000 samples with whole genome sequencing. This would complement the current analysis.

We thank the Reviewer for this suggestion as it has led us to conduct a highly interesting analysis. Briefly, we retrieved the list of somatic mutations for 3,011 tumors with whole genome sequencing included the International Cancer Genomics Consortium (ICGC, <https://icgc.org/>) and applied our method to detect regulatory elements that are mutated at a higher frequency than expected on 245,051 DHSs that are active in at least 15 of the 55 cell lines described in ENCODE. We had to modify our enrichment Filter 3 (detection of DHSs that, when mutated, are associated with aberrant expression of the genes they interact with) for the following reasons: 1) Investigating gene expression levels in multiple tissues would require normalization, which may greatly increase the noise in the detection of genes that are aberrantly expressed when their associated regulatory element is mutated; 2) only a subset of the 3,011 tumors have gene expression data available. Thus, the application of Filter 3 not feasible for the majority of DHSs. To avoid these issues, we developed a different approach: we selected DHSs that have mutations altering transcription factor binding sites (TFBS). Using this filter, we identified 121 DHSs that have at least 10 mutations altering a single TFBS. Among the genes associated with these DHSs, we found that 42 are known cancer genes², suggesting that this filter likely enriches for putative driver DHSs. In general, this pan-cancer approach would be expected to result in the loss of tissue-specific regulatory elements, as well as the loss of DHSs that harbor tumor-type specific driver mutations. Given that several of the putative driver regulatory elements

that we identified in breast cancer are tissue-specific, we did not expect to identify them as putative drivers in this pan-cancer dataset. Indeed, of the ten putative driver DHSs passing all our filters, nine were present in this analysis and only two were found significant (chr5:1325957-1328153 and chr2:216531968-216533440). For these reasons, we feel it is more appropriate to describe this pan-cancer analysis in a separate manuscript, which is currently in preparation.

3. The authors test the potential distal regulatory elements across a separate breast cancer set. What is the distribution of the specific distal regulatory elements across different breast cancer subtypes? One would expect selectivity. This should be noted in the text.

In ENCODE, two relevant breast cell lines are included: human mammary fibroblasts (HMF) and T-47D, a cell line derived from a ER+/PR+/HER2- invasive ductal carcinoma. Combining DHSs from these two cell lines likely results in the detection of the vast majority of regulatory elements active in breast tissues. Unfortunately, DHS information for additional breast cancer cell lines or tissues from different subtypes is not available.

We agree with the reviewer in that we may be missing specific distal regulatory elements that are active only in specific breast cancer subtypes. Indeed, in our significance analysis, we compared the results of the test statistic p for the mutations in breast DHSs to those derived from analyzing mutations in DHSs from 13 different control tissues (Supplementary Table 8, Supplementary Table 9). A priori we expect to see no selection signal in the control tissues and, hence, no differences between the distribution of p in the control tissues and in the simulated random data, which correspond to our model of neutral evolution. If DHSs in the control tissues were all true negatives (i.e. they do not harbor any driver mutations), the distribution of p values would be the same as in the random expectation and the FDR would be equal to one for all p . Indeed, we observe a significantly lower signal than in breast DHSs (Fig. 2C), albeit still positive, suggesting that there are DHSs active in the control tissues that may be positively selected for mutations in breast cancer. There are several possible reasons for this including the genomic regions corresponding to these DHSs may be inactive in the two breast samples that we used to define breast DHSs (HMF and T-47D), but may be active in other breast cancer samples.

We have edited the “Identifying DHSs significantly mutated in breast cancer” section in the Results on pages 5-6 to make this clearer.

4. The authors do explore one distal regulatory element near TERT in further detail. This is a positive. However, given that as they indicate in the introduction, TERT is the only gene where promoter mutations have been validated to be important in cancer, more detailed exploration of additional sites particularly the one that is present in most tumor lineages would strengthen the manuscript.

We agree with the Reviewer that a detailed exploration of additional sites would strengthen the manuscript. Following this suggestion, we performed a combination of *in silico* analyses and experimental validations that show that somatic mutations in the putative driver DHSs are functional.

We investigated the ten putative driver DHSs identified through the enrichment filters, which allowed us to determine that mutations in these elements are associated with the altered expression of known cancer genes. In particular, we found that, of the 27 aberrantly expressed genes associated with mutations in these DHSs, 18 have a known role in cancer: one is a

known cancer gene included in the Cancer Gene Census (*TRIM27*), six genes (*ACSBG1*, *COL20A1*, *DVL1*, *LPCAT1*, *VWA1*, and *ZNF596*) are aberrantly expressed in multiple cancer types and eight (*ACSBG1*, *ATAD3B*, *CLPTMIL*, *COL20A1*, *LPCAT1*, *MAST2*, *RAD54L* and *ZNF596*) are involved in breast cancer. We added this analysis to “Enrichment filters for driver DHSs in breast cancer” in the Results on pages 7-8 and created Supplementary Table 16.

In addition to the in-depth *in silico* functional characterization of the most highly mutated putative driver DHS (chr5:1325957-1328153) in breast cancer samples presented in the first submission, to address this comment we have experimentally characterized the second, third and fourth most highly mutated putative driver DHSs (chr6:28948439-28951450, chr8:579137-581436 and chr20:62115827-62119284) as described below in the response to Reviewer 1 comment 10.

5. The authors at all stages took approaches that would decrease the false negative rate and would increase FDR. For example an FDR of 125 is very liberal. This is OK but should be noted in the discussion of the data. It also warrants care in the text. “Using this threshold we identified 637 significantly mutated breast DHSs”. This statement on page 4 of the text should have a conditional. These should be listed at this point as potentially significantly mutated breast DHSs. They are not yet significantly mutated.

We agree with the Reviewer that $FDR = 25\%$ would be a loose threshold if there were no additional filters that further eliminate false positives. According to the Reviewer’s suggestion, we have modified the text in “Identifying DHSs significantly mutated in breast cancer” in the Results on page 5 to “we identified 637 potentially significantly mutated breast DHSs”.

6. The authors use “control” tissues to test the potential that the DHSs identified in breast cancer have undergone positive selection. This is a good addition to the manuscript. It would be strengthened with more samples than the 13 tissues used. Further the authors claim Indeed, we observe a significantly lower signal than in breast DHSs (Fig. 2C), albeit still positive, suggesting that there are DHSs active in the control tissues that may be positively selected for mutations in breast cancer. As indicated above, there is a second possibility which is that with a FDR of .25, there may be false positives in the normal tissue. The authors should consider or at least comment on this possibility. The subsequent arguments are reasonable.

We agree with the point made by the Reviewer about the FDR threshold used. Although the $FDR = 0.25$ filter may be loose, it should be noted that the additional filters are applied in order to decrease the number of potentially false positives that are retained using this liberal threshold.

To address the reviewers comment we have added the following sentence on page 6, “Given our use of an $FDR = 0.25$ for Filter 1, the 637 significantly mutated breast DHSs may have a high number of false positives. Therefore, to eliminate false positives and enrich for a set of DHSs with characteristics expected of driver regulatory elements, such as functional activity in associated tissue, we applied additional filters to the 637 breast DHSs.”

We used all the DHSs set that are in ENCODE, which are from 13 tissue types and 53 ENCODE cell lines (Supplementary Table 8). We think that these cover all the major human tissue-types. Additionally, using DHSs that were not generated by the ENCODE may result in batch effects that could introduce biases.

We modified text on page 6 to address both this comment and comments 3 and 5: Indeed, we observe a significantly lower signal than in breast DHSs (Fig. 2C), albeit still positive, suggesting that there are DHSs active in the control tissues that may be positively selected for mutations in breast cancer. There are several possible reasons for this including: 1) some of these DHSs are active in multiple control tissues, and a fraction of them are likely false negatives in the set of breast DHSs in ENCODE; 2) some of the significantly mutated DHSs active in control tissues may be false positives, due to the fact that we used a loose FDR threshold (0.25); 3) these DHSs may be a different class of regulatory elements in breast that do not bind DNA-binding proteins, and therefore are not detected as DHSs; and 4) the genomic regions corresponding to these DHSs may be inactive in the two breast samples that we used to define breast DHSs (HMF and T-47D), but may be active in other breast cancer samples.

7. The enrichment of potentially selected DHS at the different filter levels argues for enrichment and is a strong positive. However, it does not speak to any of the specific DHSs that are being explored. This should be noted. This is picked up in the additional filters further along.

To address this comment, we modified text on page 6:

“As has been shown in whole exome analyses to identify novel cancer genes, statistical methods that select for sequences mutated at higher frequency than expected are not sufficient to detect driver mutations and often result in the detection of false positives, such as genes like TTN or MUC17, that are neither expressed in the tumor nor in its associated normal tissue. Given our use of an FDR = 0.25 for Filter 1, the 637 significantly mutated breast DHSs may have a high number of false positives. Therefore, to eliminate false positives and enrich for a set of DHSs with characteristics expected of driver regulatory elements, such as functional activity in associated tissue, we applied additional filters to the 637 breast DHSs.”

8. The gene ontology analysis is done on an unvalidated set of DHS. This should not be done at this stage but rather after Filter 4 is performed. The associations with the 16 putative driver DHSs would be stronger.

We agree that functional enrichment analysis on the aberrantly expressed targets of the 73 DHSs was out of place. Since we added the 560 BRCA-EU breast cancer samples as part of the replication screening, the number of putative driver DHSs is now ten, which is not enough to do a functional enrichment analysis. Instead, we inspected the functions of 27 genes that have altered expression associated with mutations in these ten putative driver DHSs and established interactions (Filter 3) and found that the majority (18) have known roles in cancer. We modified the Results on page 8 and modified Supplementary Table 16 with these new data. See also the response to Comment 4 by Reviewer 1.

9. Overexpression of cancer associated genes. This is potentially the most important part of this study. It is important to emphasize that all of the studies presented are associations and not evidence of causality. One question that is not approached in this section is whether the putative driver DHS are mutually exclusive with promoter mutations for hTERT. It may be that there is not sufficient power for this comparison.

As the Reviewer suggests, we now emphasize that this association does not provide evidence of causality in the Results on page 7.

“Of note, although mutations in these DHSs are associated with aberrant expression of the target genes, this is not evidence of causality and thus we applied additional filters and conducted functional experiments as described below.”

In the greater than 650 breast cancer samples in this study with WGS, we did not observe any mutations in the *TERT* promoter nor we did not find any literature evidence of it being commonly mutated in breast cancer. We realize this was not clear in the original manuscript and have edited the text on page 9 as follows:

“Since *TERT* overexpression is often associated with mutations in its promoter, we examined the *TERT* promoter in the 1,097 TCGA Cancer Genomes (see Methods) and found 51 mutations, of which 49 are in two loci previously described as recurrently mutated (Supplementary Table 19). The majority of these 51 mutations are in glioblastoma (23) and low-grade glioma (10), with the remaining distributed across bladder, head and neck, lung, melanoma and thyroid tumors. Our findings show that the known driver mutations in the *TERT* promoter do not commonly occur in breast cancer, but *TERT* is frequently overexpressed in breast tumors and this altered expression is associated with mutations in the putative driver DHS (chr5:1325957-1328153).”

We agree with the reviewer that the possibility of mutations in the *TERT* enhancer and promoter being synergistic or exclusive is an interesting question. When investigating the mutations in the pan-cancer analysis, we found 8 tumors (5 GBM, 2 BLCA and 1 HNSC) that harbor mutations in both the promoter and enhancer. We found that *TERT* is overexpressed in these samples ($p < 1e-4$ in GBM, $p = 2e-4$ in BLCA, bootstrapping); but did not have the power to determine if there was a synergistic effect of having both elements mutated. At this point, we feel that there is insufficient data to include this analysis in the paper. Investigating additional tumors with both whole genome sequencing and gene expression data may provide more power to assess the presence of a combined effect of the mutations in *TERT* promoter and its enhancer.

10. A key question would be to determine whether introduction of the specific changes in the DHS and preferably ones that are not near hTERT actually increase the transcription of genes of interest. This was a requirement for the demonstration that the TERT promoter mutations are indeed drivers or hTERT expression.

We agree with the Reviewer that functional characterization of the putative driver DHSs is important. As described in our response to comment 4, we have added an *in silico* analysis of the ten putative driver DHSs identified through the enrichment filters, which allowed us to determine that mutations in these elements are associated with the altered expression of known cancer genes. In addition to the in-depth *in silico* functional characterization of the most highly mutated putative driver DHS (chr5:1325957-1328153) in breast cancer samples presented in the first submission, we have experimentally characterized the second, third and fourth most highly mutated putative driver DHSs. These experimental validations enabled us to show that: 1) mutations in putative driver DHS chr6:28948439-28951450 affect the regulatory potential of the element; and 2) putative driver DHSs chr8:579137-581436 and chr20:62115827-62119284 control the expression levels of the target genes identified in Filter 3, several of which are known cancer genes. Specifically, we conducted the following experiments:

1. Mutations in the second most highly mutated putative driver in breast cancer, DHS chr6:28948439-28951450 (Fig. 3A), were associated in Filter 3 with the overexpression of a known cancer gene (*TRIM27*). To determine if the mutations we identified in this DHS have a functional impact on gene expression, and therefore may be driver mutations, we investigated the effects of four mutations *in vivo*, using *Ciona intestinalis* as a model system. This urochordate is an excellent system to use for screening of regulatory variants, because it shares a large part of its transcriptional machinery with higher eukaryotes³. The four mutations are distributed across the element with two affecting GATA binding sites and two affecting ETS binding sites). To determine their effects, we built reporter constructs containing either the wild-type or mutated DHS attached to a minimal promoter⁴ and GFP. These constructions were electroporated into *C. intestinalis* fertilized eggs to assay their function. The two GATA mutations result in significant differential expression, with one resulting in overexpression and one in downregulation (p-value = $1.25 \cdot 10^{-12}$ and p-value = 0.0032, respectively Fisher's exact test): chr6:28950885A>G results in a 7-fold increased GFP signal in epidermis, whereas chr6:28949254A>C reduces the enhancer activity by 50% in anterior neural plate (a6.5 lineage). We also observed a significant decrease in the enhancer activity for chr6:28950050G>A in multiple tissues (p-value = 0.0077 in endoderm, p-value = 0.065 in anterior neural plate and p-value = 0.015 in secondary notochord), whereas chr6:28950040C>T does not result in any significant change. These results demonstrate that three of the four mutations tested result in the aberrant activity of DHS chr6:28948439-28951450, providing evidence that these were likely driver mutations in the cancers in which they were detected. We added new sections to describe these results "Mutations in DHS chr6:28948439-28951450 affect regulatory potential" on pages 9-10 and "*In vivo* analysis of the effects of mutations in putative driver DHS chr6:28948439-28951450" in the Methods on page 21, Fig. 5 and Supplementary Tables 20, 21, 24.
2. In Filter 3 (Fig. 2A) we found associations between mutations in the putative driver DHSs (chr8:579137-581436 and chr20:62115827-62119284) and altered expression of target genes. We deleted DHSs chr8:579137-581436 and chr20:62115827-62119284 to determine if they control the expression levels of these target genes, providing evidence that the associated mutations are potentially causative. We used CRISPR to delete the two DHSs and performed RNA-seq and ATAC-seq to detect changes in gene expression and chromatin accessibility upon the deletion of the two DHSs. The deletion of both putative driver DHSs results in the significant downregulation of several genes and a remodeling of the chromatin in the promoters of several genes, including the target genes identified in Filter 3. We added a new section describing these results "Long-range targets of putative driver DHSs" on page 10-11 and "Deletion of driver DHSs (chr8:579137-581436 and chr20:62115827-62119284) in HEK293T cell line" in the Methods on pages 21-23 and created Figure 6, Supplementary Fig. 6, 11, 12, 13, Supplementary Tables 25-28. See also comment 4 by Reviewer #1 and comment 4 by Reviewer #2.

Reviewer #2 (Remarks to the Author):

This paper describes a strategy for identifying candidate driver mutations in breast cancers using whole genome sequences from tumor samples. The main features are to (a) focus on candidate regulatory regions using chromatin accessibility data (DNase hypersensitive sites or DHSs) measured in relevant cell types and (b) identify DHSs with mutation rates in the tumors that are significantly higher than expected under a neutral mutation model. The authors apply several filters to focus on DHSs containing mutations more likely to be drivers of tumors. One of the three DHSs containing likely driver mutations was shown to be associated with genes with altered levels of expression in tumors, leading to the hypothesis that the candidate driver mutations in this one DHS are affecting expression levels of several genes, including the gene TERT, which has been previously implicated in tumors.

The novel part of this report is developing a method for dealing with the five known genomic and epigenomic features that affect the local rate of mutations, specifically “1) DNA replication timing, 2) open and closed chromatin status, 3) GC content, 4) local gene density, and 5) expected mutations based on trinucleotide composition”. The authors annotated almost 335,000 breast DHSs by these five features, and used k-means clustering to identify 223 clusters of DHSs similar in these features, with more than 100 DHSs per cluster. They inferred the background mutation probability for each cluster and identified DHSs that were mutated in breast cancer at a rate significantly higher than expected under a neutral model. The method developed to do this is a notable advance because it takes into account the several features that impact local rates of mutation, and it does so in a data-driven manner. The authors make a strong case that their estimates of enrichment and significance are conservative. Appropriate statistical tests are applied.

Identifying strong candidates for mutations or variants in regulatory regions that affect phenotypes is currently a topic of high interest. Many investigators are taking approaches largely examining epigenetic features such as histone modifications or transcription factor occupancy. While those approaches have high value, the current report shows the continuing value of rigorous searches for evidence of positive selection as a complementary guide for finding functionally significant mutations.

The report could be improved by addressing the following issues.

(1) The authors open the section on finding DHSs significantly mutated in breast cancer (page 4) with the comment that “the local mutation density is lower in regulatory elements that are active in the cell-type of origin of the tumor compared to regulatory elements in other cell types”. Thus it may be puzzling to readers that the focus is on finding DHSs with a higher frequency of somatic mutations as an indicator of drivers. The answer is given in the Methods: “The signature of positive selection for mutations at a driver locus in the tumor genome is an increase in observed mutation density relative to the expected density under a neutral model of evolution.” That explanation should also be stated clearly in the Results.

We agree with the Reviewer that this could be presently clearer. To respond to this comment, we edited the Results on page 5 as follows: “Several groups have previously shown that the local mutation density is lower in regulatory elements that are active in the cell-type of origin of the tumor compared to regulatory elements in other cell types. This observation supports the hypothesis that mutations in functional genomic regions are usually deleterious and negatively selected. Therefore, increased local mutation density relative to the expected density under a neutral model of evolution is evidence of positive selection, suggesting that the mutations are functional and hence referred to as drivers.”

(2) After identifying 16 putative driver DHSs associated with breast cancer (top of page 6), the authors then focus on three DHSs with elevated mutation rates in multiple cancers. This leaves the impression that the other 13 putative breast cancer driver DHSs are not as important. The authors should clarify how readers should interpret these different sets of putative driver DHSs.

We thank the reviewer for bringing this point to our attention. We did not intend to suggest that only the three DHSs mutated in the pan-cancer analysis are drivers. We have modified the structure of the manuscript to deal with this inadvertent implication that only these three DHSs are drivers in breast cancer.

Specific modifications include:

1. Based on an additional analysis requested by Reviewer 1, there are now ten putative driver DHSs. We characterize the target genes of all ten putative driver DHSs for their role in cancer as described in our response to Reviewer 1 comment 4.
2. We have changed the focus in the Results to validation of four breast cancer putative driver DHSs (experiments conducted in response to Reviewer 1 comment 10) rather than on the pan-cancer analysis. The pan-cancer analysis is now one of several analyses conducted to further characterize the breast cancer putative driver DHSs.
3. We added the following sentence in the Discussion on pages 11-12, “Although we did not experimentally investigate the other six putative breast cancer driver DHSs, the fact that mutations in many of them are associated with altered expression of genes with known roles in breast cancer suggests that they may be breast cancer-specific drivers.”

(3) While the authors focus on the putative driver DHS in an intron of the gene CLPTMIL (Figure 4, the other two pan-cancer driver DHSs are not discussed. One of them, chr8:579137-581436, is located close to a telomere. Other studies have shown a high rate of substitution close to telomeres (e.g. doi: 10.1073/pnas.1221792110). Could this chromosomal location contribute to the observed elevated mutation rate?

We understand why this may be of concern to the reviewer, given that mutation rates at telomeres are higher. However, we addressed the potential confounding effect of local mutation rates in the pan-cancer analysis of the ten putative driver DHSs. In the interval analysis described on page 8, we investigated the mutation rates of the 100 kb intervals surrounding each of the ten DHSs (Fig. 3C) and found that three are significantly more mutated than their surrounding genomic regions, including chr8:579137-581436 and chr20:62115827-62119284, which are both located close to the telomere (Fig. 3C-D). Furthermore, as described in our response to Reviewer 1 comment 10, we added an *in vitro* experimental validation of these two putative driver DHSs, demonstrating that they causally influence the expression of several cancer genes identified in Filter 3. These results show that, although these two DHSs are located close to telomeres, their significantly high mutation rate is likely evidence of positive selection.

(4) The third DHS, chr20:62115827-62119284, does not appear to be a strong DHS. More discussion of these exemplar candidate driver DHSs, or alternatively, other breast cancer candidate drivers, would help readers appreciate the value and perhaps limitations of the results.

The peaks associated with DHS chr20:62115827-62119284 in the two breast tissues HMF and T-47D are not as strong as the other nine putative driver DHSs. However, DHS chr20:62115827-62119284 is not only significantly mutated in breast cancer but was also significantly mutated in the pan-cancer analysis as well as seven of the other cancer types individually (Supplementary Table 17). Therefore, the relationship between DHS peak intensity and driver potential is not clear. Additionally, as described in response to Reviewer 1 comment 10, we deleted the putative driver DHS chr20:62115827-62119284 in a normal cell line using CRISPR and investigated the effects of its deletion on gene expression (RNA-seq). We found that several genes are downregulated when this region is deleted (including the known cancer gene *ARFGAP1* identified in Filter 3), indicating that it indeed is a regulatory element.

(5) page 4, top line, comments twice on the gene MAP3K1, but seemingly in two distinct groups; this is confusing.

We thank the Reviewer for pointing this mistake out. We have corrected this: among the known driver genes in breast cancer *MAP2K4* is recurrently mutated at lower frequency, while *MAP3K1* is mutated at high frequency.

(6) Fig. 2A: Filter 3 recovers 73 DHSs from an initial 596, but only 7 are shown as being removed. Is this an error? If not, where did the other DHSs go?

We thank the Reviewer for pointing this mistake out. We have corrected this error in Fig. 2A: the number of DHSs eliminated with Filter 3 is 523.

(7) Fig. 3A and 3B have no labels on the axes.

We have added the labels to Fig. 3A and B.

(8) Fig. 3C would be easier to understand if the description in the Results used the same phrases as the labels on the axes, i.e. clusters analyses and interval analyses.

We thank the Reviewer for pointing this out. We have added the definition “clusters analysis” and “interval analysis” to the descriptions of the analysis in the Results on page 8.

References

- 1 Nik-Zainal, S. *et al.* Landscape of somatic mutations in 560 breast cancer whole-genome sequences. *Nature* **534**, 47-54, doi:10.1038/nature17676 (2016).
- 2 Futreal, P. A. *et al.* A census of human cancer genes. *Nature reviews. Cancer* **4**, 177-183, doi:10.1038/nrc1299 (2004).
- 3 Farley, E. K. *et al.* Suboptimization of developmental enhancers. *Science* **350**, 325-328, doi:10.1126/science.aac6948 (2015).
- 4 Rothbacher, U., Bertrand, V., Lamy, C. & Lemaire, P. A combinatorial code of maternal GATA, Ets and beta-catenin-TCF transcription factors specifies and patterns the early ascidian ectoderm. *Development* **134**, 4023-4032, doi:10.1242/dev.010850 (2007).

REVIEWERS' COMMENTS:

Reviewer #1 (Remarks to the Author):

The authors have significantly revised the manuscript and have included both a new large sample set and have validated experimentally a number of candidate DHS. Thus this manuscript is strengthened and my most of my key concerns have been resolved.

There remain a series of minor concerns that need to be revised.

In the introduction the authors note

“In this study, we have analyzed whole genome sequences for 97 breast cancer samples¹¹ and more than one thousand tumors across 19 additional cancer types to detect non-coding driver mutations.”

I think this needs to be updated to fit the addition of the 560 new breast cancer samples.

The authors need to emphasize that there is no relationship between mammary fibroblasts and ductal breast cancers other than they are from the same anatomic location. Thus they are not particularly relevant to human breast cancer as is stated in the manuscript. The fibroblasts are not in the same lineage as the epithelial cells that lead to ductal carcinoma. This needs to be corrected in the results section and mentioned in the discussion.

The authors have still not fully dealt with the fact that breast cancers likely have multiple tissues of origin that need to be factored in to the analysis. Please add to the discussion that an analysis starting with multiple cell lines representing the diversity of breast cancer would likely alter the candidates identified.

Reviewer #2 (Remarks to the Author):

The authors have clarified the previously unclear points. Importantly, they have performed new experimental tests showing an impact of the candidate driver mutations on expression (transfections of Ciona) or a change in chromatin accessibility and expression upon deletion of the DHSs. The latter support the activity of the candidate enhancers, and while they do not show an effect of the candidate driver mutations, they support the fundamental hypothesis that the authors' method is identifying mutations that are strong candidates for affecting gene expression.

Reviewer: Ross Hardison

REVIEWERS' COMMENTS:

Reviewer #1 (Remarks to the Author):

The authors have significantly revised the manuscript and have included both a new large sample set and have validated experimentally a number of candidate DHS. Thus this manuscript is strengthened and my most of my key concerns have been resolved.

There remain a series of minor concerns that need to be revised.

1. In the introduction the authors note

“In this study, we have analyzed whole genome sequences for 97 breast cancer samples¹¹ and more than one thousand tumors across 19 additional cancer types to detect non-coding driver mutations.”

I think this needs to be updated to fit the addition of the 560 new breast cancer samples.

We have changed this number to 657.

2. The authors need to emphasize that there is no relationship between mammary fibroblasts and ductal breast cancers other than they are from the same anatomic location. Thus they are not particularly relevant to human breast cancer as is stated in the manuscript. The fibroblasts are not in the same lineage as the epithelial cells that lead to ductal carcinoma. This needs to be corrected in the results section and mentioned in the discussion.

To respond to the reviewer's comment, we have added a sentence in the Discussion:

We analyzed DHSs that are active in breast (derived from HMF and T47-D cell lines) because it was shown that chromatin features in the tissue of origin of a tumor are strongly associated with the tumor's somatic mutation profile

3. The authors have still not fully dealt with the fact that breast cancers likely have multiple tissues of origin that need to be factored in to the analysis. Please add to the discussion that an analysis starting with multiple cell lines representing the diversity of breast cancer would likely alter the candidates identified.

We have discussed why we used this approach in the Introduction:

Mutational analyses of exomes have shown that combining all breast cancer subtypes together results in increased sensitivity for identifying driver genes.

However we realize that this is an important point and have added two sentences in the Discussion:

We combined all breast cancer subtypes together because previous studies have shown that this solution results in increased sensitivity for identifying driver genes. However, analyzing the different subtypes separately may result in the detection of additional subtype-specific drivers.

Reviewer #2 (Remarks to the Author):

The authors have clarified the previously unclear points. Importantly, they have performed new experimental tests showing an impact of the candidate driver mutations on expression (transfections of Ciona) or a change in chromatin accessibility and expression upon deletion of the DHSs. The latter support the activity of the candidate enhancers, and

while they do not show an effect of the candidate driver mutations, they support the fundamental hypothesis that the authors' method is identifying mutations that are strong candidates for affecting gene expression.

Reviewer: Ross Hardison